# Trajectory Stitching for Solving Inverse Problems with Flow-Based Models

**Alexander Denker**[† 1]  **Zeljko Kereta**[2]  **Carola-Bibiane Schönlieb**[3]  **Moshe Eliasof**[3 4]

## Abstract

Flow-based generative models have emerged as powerful priors for solving inverse problems. One option is to directly optimize the initial latent code (noise), such that the flow output solves the inverse problem. However, this requires backpropagating through the entire generative trajectory, incurring high memory costs and numerical instability. We propose MS-Flow, which represents the trajectory as a sequence of intermediate latent states rather than a single initial code. By enforcing the flow dynamics locally and coupling segments through trajectory-matching penalties, MS-Flow alternates between updating intermediate latent states and enforcing consistency with observed data. This reduces memory consumption while improving reconstruction quality. We demonstrate the effectiveness of MS-Flow over existing methods on image recovery and inverse problems, including inpainting, super-resolution, and computed tomography.

## 1. Introduction

Inverse problems in imaging, such as super-resolution, inpainting, and compressed sensing, aim to recover a high-fidelity signal $\mathbf{x} \in \mathbb{R}^n$ from noisy, indirect measurements $\mathbf{y} \in \mathbb{R}^m$. This process is typically modeled by the forward equation $\mathbf{y} = \mathbf{A}\mathbf{x} + \boldsymbol{\eta}$, where $\mathbf{A}$ is a known forward operator and $\boldsymbol{\eta}$ represents measurement noise. Due to the ill-posed nature of this problem, relying solely on data consistency is insufficient, and regularization is required to constrain the solution space to valid images (Engl et al., 1996).

Deep generative models have emerged as powerful regular-

---

[†]Work done while at University College London. [1]Helmholtz Imaging, Deutsches Elektronen-Synchrotron DESY, Germany [2]Department of Computer Science, University College London [3]Department of Applied Mathematics and Theoretical Physics, University of Cambridge [4]Faculty of Computer and Information Science, Ben-Gurion University of the Negev. Correspondence to: Alexander Denker <a.denker@ucl.ac.uk>.

*Proceedings of the $43^{rd}$ International Conference on Machine Learning*, Seoul, South Korea. PMLR 306, 2026. Copyright 2026 by the author(s).

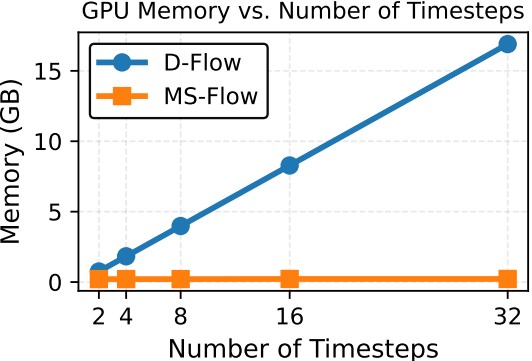

*Figure 1.* Peak GPU memory for D-Flow vs MS-Flow (Ours) using Euler discretization of the ODE on CelebA. D-Flow scales linearly with the number of timesteps, while MS-Flow is constant.

ization methods (Duff et al., 2024). By learning a mapping $g_\theta : \mathbb{R}^n \to \mathbb{R}^n$ that pushes a simple latent distribution $p_{\mathbf{z}}$ (e.g., $\mathcal{N}(0, \mathbf{I})$) to a complex data distribution $p_{\mathbf{x}}$, these models act as powerful, learned priors. A common strategy for leveraging these priors is Latent Space Optimization (LSO) (Bora et al., 2017; Menon et al., 2020). LSO seeks a reconstruction $\hat{\mathbf{x}} = g_\theta(\hat{\mathbf{z}})$ by optimizing the latent code via

$$\hat{\mathbf{z}} \in \arg \min_{\mathbf{z} \in \mathbb{R}^n} \Phi(g_\theta(\mathbf{z})) + \lambda \mathcal{R}(\mathbf{z}), \qquad (1)$$

where $\Phi$ enforces data consistency (e.g., $\Phi(\mathbf{x}) = \frac{1}{2}\|\mathbf{A}\mathbf{x} - \mathbf{y}\|^2$ under Gaussian noise), and $\mathcal{R}$ acts as a regularizer for the latent code, ensuring that $\mathbf{z}$ remains within high-density regions of the latent space.

LSO has achieved remarkable success in imaging inverse problems. Key examples are super-resolution (Menon et al., 2020), compressed sensing (Bora et al., 2017), or inpainting (Asim et al., 2020), leveraging a range of deep generative models, e.g., normalizing flows (Rezende & Mohamed, 2015), variational autoencoders (Kingma & Welling, 2013), or generative adversarial networks (Goodfellow et al., 2014), as backbones. LSO has also been used in conjunction with diffusion models (Wang et al., 2024a; Chihaoui et al., 2024), see also related work in Appendix A. Importantly, LSO is not limited to inverse problems but also underlies tasks such as classifier-guided or conditional sampling, where $\Phi$ encodes semantic constraints rather than data-consistency.

In continuous-time generative models, such as continuous normalizing flows (Chen et al., 2018; Grathwohl et al., 2019)

or flow-based models (Lipman et al., 2023), the generator $g_\theta$ is not given in closed form. Instead, samples are defined as the solutions of an ordinary differential equation (ODE)

$$\frac{d\mathbf{x}(t)}{dt} = v_\theta(\mathbf{x}(t), t), \qquad \mathbf{x}(0) = \mathbf{z}, \quad t \in [0, 1], \quad (2)$$

where $v_\theta$ is a trained model. Substituting Equation (2) into Equation (1) turns LSO into an ODE-constrained optimization problem (Lions, 1971; Hinze et al., 2008). The current state of the art, D-Flow (Ben-Hamu et al., 2024), tackles LSO via a *single-shooting* strategy, in which the ODE is discretized, and gradients are backpropagated through the resulting trajectory. While effective, this approach suffers from two critical limitations:

**(L1) High Memory Cost.** Backpropagation through the full ODE trajectory requires storing, or recomputing, intermediate states. Checkpointing (Chen et al., 2016) trades memory for compute, while adjoint methods (Chen et al., 2018) require solving an additional backward ODE, increasing the computational cost.

**(L2) Poor Conditioning.** The generated image $\mathbf{x}(1)$ depends nonlinearly on the initial latent code $\mathbf{z} = \mathbf{x}(0)$. Optimizing through a long composition of neural networks can cause vanishing or exploding gradients, degrading convergence and reconstruction quality.

To overcome these limitations, we propose MS-Flow, a *multiple-shooting* framework for LSO with flow-based generative models. MS-Flow decomposes the ODE trajectory into short segments and replaces the hard global ODE constraint with soft stitching penalties. This yields:

1. **Decoupled Optimization.** We employ an alternating minimization strategy that separates the neural dynamics from data-consistency updates.
2. **Reduced Memory Footprint.** Optimization avoids backpropagation through the full time horizon, resulting in memory usage that is constant with respect to the ODE discretization.

Moreover, we introduce a Jacobian-free update for trajectory variables, further reducing memory and computational cost, resulting in a fast, scalable and stable method. See Figure 1 and Section 5 for details. Further, we compare with other recently proposed inverse problem solvers making use of flow models. We discuss related works in Appendix A.

## 2. Background and Preliminaries

In this section, we provide an overview of the definitions and methodologies used throughout this paper.

### 2.1. Flow-based Models

Flow Matching (FM) (Lipman et al., 2023) trains continuous normalizing flows using a simulation-free objective. In

FM, a time-dependent vector field $v_\theta : \mathbb{R}^n \times [0, 1] \to \mathbb{R}^n$ that induces a probability path $p_t$ transporting a simple base distribution, e.g., a standard Gaussian, to the data distribution $p_1$ is learned. The training objective is to align $v_\theta$ with a ground-truth velocity field $u_t$ that generates the target path $p_t$, leading to the flow matching objective

$$\mathcal{L}_{\mathrm{FM}}(\theta) = \mathbb{E}_{t, \mathbf{x} \sim p_t(\mathbf{x})} \| v_\theta(\mathbf{x}, t) - u_t(\mathbf{x}) \|^2. \quad (3)$$

However, as the evaluation of the marginal velocity $u_t$ is generally intractable, FM adopts a conditional formulation. Conditional FM training matches a conditional velocity field

$$\mathcal{L}_{\mathrm{CFM}}(\theta) = \mathbb{E}_{t, \mathbf{x}_0, \mathbf{x}_1,} \| v_\theta(\mathbf{x}_t, t) - u_t(\mathbf{x}_t \mid \mathbf{x}_0, \mathbf{x}_1) \|^2, \quad (4)$$

with $\mathbf{x}_t \sim p_t(\mathbf{x}_t \mid \mathbf{x}_0, \mathbf{x}_1)$. The gradients of the conditional and marginal objectives coincide, i.e., $\nabla_\theta \mathcal{L}_{\mathrm{CFM}}(\theta) = \nabla_\theta \mathcal{L}_{\mathrm{FM}}(\theta)$ (Lipman et al., 2023; Tong et al., 2023). Sampling is obtained by simulating the ODE in Equation (2). In practice, the conditional path is often defined via affine interpolation between a source sample $\mathbf{x}_0 \sim p_0$ and a target data point $\mathbf{x}_1 \sim p_1$, e.g., with an optimal transport displacement $\mathbf{x}_t = (1 - t)\mathbf{x}_0 + t\mathbf{x}_1$ (Liu, 2022), which yields a constant conditional velocity $u_t(\mathbf{x}_t | \mathbf{x}_0, \mathbf{x}_1) = \mathbf{x}_1 - \mathbf{x}_0$. With $\mathbf{x}_t = (1 - t)\mathbf{x}_0 + t\mathbf{x}_1$, this simplifies the CFM objective to

$$\mathcal{L}_{\mathrm{CFM}}(\theta) = \mathbb{E}_{t, \mathbf{x}_0, \mathbf{x}_1,} \| v_\theta(\mathbf{x}_t, t) - (\mathbf{x}_1 - \mathbf{x}_0) \|^2. \quad (5)$$

### 2.2. ODE-constrained Optimization

Many optimization problems in control, inverse problems, and machine learning involve objectives that depend on the solution of a dynamical system. A standard form of ODE-constrained optimization is

$$\min_u \Phi(\mathbf{x}(T), u), \, \mathrm{s.t.} \frac{d\mathbf{x}(t)}{dt} = v(\mathbf{x}(t), t, u), \mathbf{x}(0) = \mathbf{x}_0$$

where $t \in [0, T]$, $\mathbf{x}(t) \in \mathbb{R}^n$ is the system state, and $u$ represents controls, parameters, or initial conditions (Lions, 1971; Hinze et al., 2008). In flow-based generative models, the dynamics $v_\theta$ are learned and the initial condition $\mathbf{x}(0) = \mathbf{z}$ is optimized. Since the objective depends on the terminal state $\mathbf{x}(T)$, gradient computation is inherently coupled to ODE integration. Two widely used frameworks for addressing this class of problems are:

**Single-Shooting** enforces the dynamics exactly by parametrizing the entire trajectory through $\mathbf{x}(0)$. While conceptually simple, it is memory-intensive and often poorly conditioned for nonlinear dynamics (Betts, 2010).

**Multiple-Shooting** lifts the problem by introducing intermediate states $\{\mathbf{x}_k\}_{k=0}^K$ at time points $\{t_k\}_{k=0}^K$ (Bock & Plitt, 1984). Dynamics are enforced locally on each subinterval, while continuity is imposed via constraints or penalties. This breaks a long trajectory into shorter segments, improves conditioning, and enables structured optimization methods such as alternating minimization or block coordinate descent, without differentiating through the full time horizon.

## 3. Multiple-Shooting Flow

We now present MS-Flow, our multiple-shooting approach for solving inverse problems with flow-based models.

### 3.1. From Single- to Multiple-Shooting

Existing approaches, such as D-Flow (Ben-Hamu et al., 2024) or DMPlug (Wang et al., 2024a) for diffusion models, consider a *single-shooting* approach and aim to solve the constrained optimization problem

$$\min_{\mathbf{x}_0} \Phi(\mathbf{x}(1)), \text{ s.t. } \frac{d\mathbf{x}}{dt} = v_\theta(\mathbf{x}, t), t \in [0, 1], \mathbf{x}(0) = \mathbf{x}_0, \quad (6)$$

where $v_\theta$ is a pre-trained (frozen) flow matching model. In particular, the optimization is done with respect to the initial value of the ODE, i.e., $\mathbf{x}(0)$. As discussed in Section 1, while viable, this approach comes with a high memory cost because it requires gradients to be backpropagated through the full discretized trajectory, and in particular through $v_\theta$.

To address these limitations, we adopt the multiple shooting principle (Bock & Plitt, 1984; Wirsching et al., 2007). We partition the time interval $[0, 1]$ into $K$ segments with grid points $0 = t_0 < t_1 < \cdots < t_K = 1$. That is, instead of optimizing a single trajectory generated from $\mathbf{x}_0$, we optimize over a collection of decision variables $\{\mathbf{x}_0, \ldots, \mathbf{x}_K\}$, representing intermediate states, also referred to as shooting points, along the trajectory. Therefore, we call our approach MS-Flow, whose objective reads

$$\min_{\mathbf{x}_*, \mathbf{x}_{0:K}} \underbrace{\Phi(\mathbf{x}_*)}_{\text{Data Consistency}} + \underbrace{\lambda\mathcal{R}(\mathbf{x}_0) + \frac{\alpha}{2}\|\mathbf{x}_* - \mathbf{x}_K\|^2}_{\text{Regularization}}$$
$$+ \underbrace{\frac{\gamma}{2}\sum_{k=1}^{K}\|\mathbf{x}_k - F_{k-1,k}(\mathbf{x}_{k-1})\|^2}_{\text{Trajectory Consistency}}, \quad (7)$$

where $F_{k-1,k}(\mathbf{x}) = \texttt{odeint}(\mathbf{x}, v_\theta, [t_{k-1}, t_k])$ denotes the ODE integration from time $t_{k-1}$ to $t_k$ starting from state $\mathbf{x}$. This reformulation decouples optimization from exact ODE feasibility at every iteration. By relaxing trajectory continuity through quadratic penalties, the optimization no longer requires differentiating through the full time horizon, improving numerical conditioning (Wright et al., 1999). Moreover, the auxiliary variable $\mathbf{x}_*$ further separates data-consistency from trajectory integration, enabling the terminal state to deviate from the ODE solution and providing a trade-off between adherence to the dynamics and data consistency.

The MS-Flow objective can also be interpreted as MAP estimation over a joint posterior $p(\mathbf{x}_*, \mathbf{x}_{0:K} \mid y)$, see Appendix G. In particular, this allows us to extend the objective to diffusion models (Ho et al., 2020).

Overall, MS-Flow decomposes the optimization problem into the following three components.

**(i) Data Consistency.** The term $\Phi(\mathbf{x}_*)$ encodes the desired behavior of the model output $\mathbf{x}_*$. In the context of inverse problems we choose $\Phi(\mathbf{x}) = \frac{1}{2}\|\mathbf{A}\mathbf{x} - \mathbf{y}\|^2$, enforcing data consistency.

**(ii) Regularization.** The coupling term $\|\mathbf{x}_* - \mathbf{x}_K\|^2$ encourages the predicted inverse problem solution $\mathbf{x}_*$ to lie near the manifold of images generated by the flow model. The initial point regularizer $\mathcal{R}(\mathbf{x}_0)$ encourages the trajectory to start from high-probability regions of the latent distribution. A common choice is the negative log-likelihood of the base distribution, $\mathcal{R}(\mathbf{x}_0) = -\log p(\mathbf{x}_0)$. For a standard Gaussian, this yields $\mathcal{R}(\mathbf{x}_0) = \frac{1}{2}\|\mathbf{x}_0\|^2$ (Bora et al., 2017). However, this regularizer concentrates all the mass at the origin, even though typical samples from a Gaussian lie at a radius $O(\sqrt{d})$, see, e.g., (Bishop & Nasrabadi, 2006). We instead employ a radial Gaussian prior (Farquhar et al., 2020; Samuel et al., 2023; Ben-Hamu et al., 2024), in which the radius $r = \|\mathbf{x}_0\|$ follows a $\chi^d$ distribution. The corresponding negative log-likelihood is

$$\mathcal{R}(\mathbf{x}_0) = \frac{\|\mathbf{x}_0\|^2}{2} - (d-1)\log\|\mathbf{x}_0\| + c, \quad (8)$$

where $c$ is a constant independent of $\mathbf{x}_0$. This choice aligns the regularizer with the typical set of the latent distribution, so that the optimization is not biased to atypical latent codes.

**(iii) Trajectory Consistency.** The trajectory penalty enforces continuity between shooting points, ensuring consistency with the underlying flow dynamics.

Finally, for practical implementation, we employ an explicit Euler step for the discretization of $F_{k-1,k}$ with step size $\Delta_k = t_k - t_{k-1}$, that is further discussed in Appendix B. In this case, the objective simplifies to

$$\min_{\mathbf{x}_*, \mathbf{x}_{0:K}} \Phi(\mathbf{x}_*) + \frac{\alpha}{2}\|\mathbf{x}_* - \mathbf{x}_K\|^2 + \lambda\mathcal{R}(\mathbf{x}_0)$$
$$+ \frac{\gamma}{2}\sum_{k=1}^{K}\|\mathbf{x}_k - [\mathbf{x}_{k-1} + \Delta_k v_\theta(\mathbf{x}_{k-1}, t_{k-1})]\|^2. \quad (9)$$

### 3.2. Optimizing the MS-Flow Objective

In this section, we discuss the optimization of our MS-Flow objective in Equation (7), which lends itself to an alternating minimization strategy (Wright et al., 1999) that decouples the trajectory consistency from the data consistency. That is, we iterate between solving for the shooting points $X^i := [\mathbf{x}_0^i, \ldots, \mathbf{x}_K^i]$ and for the final inverse problem

solution estimate $\mathbf{x}_*^i$, where $i$ is the iteration index, as

$$X^{i+1} = \underset{[\mathbf{x}_0,\ldots,\mathbf{x}_K]}{\arg\min} \frac{\alpha}{2}\|\mathbf{x}_*^i - \mathbf{x}_K\|^2 + \lambda\mathcal{R}(\mathbf{x}_0)$$
$$+ \frac{\gamma}{2}\sum_{k=1}^{K}\|\mathbf{x}_k - F_{k-1,k}(\mathbf{x}_{k-1})\|^2, \quad (10a)$$

$$\mathbf{x}_*^{i+1} = \underset{\mathbf{x}_*}{\arg\min} \Phi(\mathbf{x}_*) + \frac{\alpha}{2}\|\mathbf{x}_* - \mathbf{x}_K^{i+1}\|^2. \quad (10b)$$

The overall procedure for optimizing these problems is summarized in Algorithm 1. In what follows, we provide details and discussions on the different components in Algorithm 1.

**Solving Trajectory Consistency (Equation (10a)).** Define

$$\mathcal{J}(\mathbf{x}_0,\ldots,\mathbf{x}_K) := \frac{\alpha}{2}\|\mathbf{x}_*^i - \mathbf{x}_K\|^2 + \lambda\mathcal{R}(\mathbf{x}_0)$$
$$+ \frac{\gamma}{2}\sum_{k=1}^{K}\|\mathbf{x}_k - F_{k-1,k}(\mathbf{x}_{k-1})\|^2. \quad (11)$$

We see that the gradient of Equation (11) with respect to $\mathbf{x}_k$ depends only on the intermediate neighbors $\mathbf{x}_{k-1}$ and $\mathbf{x}_{k+1}$. This motivates using coordinate descent, a common choice in such cases (Wright et al., 1999). That is, we perform a *backward sweep* from the terminal point $\mathbf{x}_K$ to the initial point $\mathbf{x}_0$, iteratively updating each shooting point $\mathbf{x}_k$ with a gradient descent step with step size $\eta > 0$, and using the most recently updated neighbors, as described in Lines 6–14 in Algorithm 1. When the transition maps $F_{k,k+1}$ are discretized using explicit Euler with step size $\Delta_k = t_{k+1} - t_k$, the coordinate-descent updates involve vector–Jacobian products (VJPs) with:

$$J_{F_{k,k+1}}(\mathbf{x}) = I + \Delta_k J_{v_\theta(\mathbf{x},t_k)}. \quad (12)$$

To avoid back-propagation of the flow model, we use a *Jacobian-free approximation* via $J_{F_{k,k+1}}(\mathbf{x}) \approx I$. We analyze the convergence properties of this choice in Section 4.2.

**Solving Data Consistency (Equation (10b)).** The data consistency step (Lines 15–17 in Algorithm 1) consists of solving a Tikhonov-regularized optimization problem that balances the objective $\Phi$ with proximity to the current shooting point $\mathbf{x}_K^{i+1}$. We can consider two special cases:

1. If $\Phi$ is a closed proper convex function, we can identify Equation (10b) with the proximal operator (Parikh et al., 2014). In particular, this allows us to apply MS-Flow to non-smooth convex objective functions, e.g., those arising in inverse problems with salt-and-pepper or impulse noise (Nikolova, 2004).

2. For linear inverse problems with additive noise, the common choice is $\Phi(\mathbf{x}) = \frac{1}{2}\|\mathbf{A}\mathbf{x} - \mathbf{y}\|^2$, and the solution of Equation (10b) is given in explicit form as

$$\mathbf{x}_*^{i+1} = (\mathbf{A}^T\mathbf{A} + \alpha I)^{-1}(\mathbf{A}^T\mathbf{y} + \alpha\mathbf{x}_K^{i+1}). \quad (13)$$

---

**Algorithm 1** Alternating Minimization with Coordinate Descent Trajectory Optimization of the MS-Flow Objective

1: **Input:** forward operator $\mathbf{A}$, data $\mathbf{y}$, flow model $v_\theta$, hyperparameters $\alpha, \gamma, \lambda$, number of trajectory update steps $L$, step size $\eta$.
2: **Initialize:** Initial inverse solution $\mathbf{x}_*^0$, initial trajectory sequence $X^0 = [\mathbf{x}_0^0, \ldots, \mathbf{x}_K^0]$.
3: $i \leftarrow 0$
4: **while** not converged **do**
5:     *// 1. Backward Sweep of Trajectory Coordinates*
6:     Initialize trajectory iterate $X^{(\ell=0)} \leftarrow X^i$
7:     **for** $\ell = 1$ to $L$ **do**
8:         Update $\mathbf{x}_K^{(\ell)}$ using $\mathbf{x}_*^i$ and $\mathbf{x}_{K-1}^{(\ell-1)}$ (Eq. 23)
9:         **for** $k = K-1$ down to 1 **do**
10:           Update $\mathbf{x}_k^{(\ell)}$ using $\mathbf{x}_{k-1}^{(\ell-1)}$ and $\mathbf{x}_{k+1}^{(\ell)}$ (Eq. 24)
11:         **end for**
12:         Update $\mathbf{x}_0^{(\ell)}$ using $\mathbf{x}_1^{(\ell)}$ (Eq. 25)
13:     **end for**
14:     Set $X^{i+1} \leftarrow X^{(L)}$
15:     *// 2. Data Consistency Optimization*
16:     Update inverse solution estimate:

$$\mathbf{x}_*^{i+1} = \underset{\mathbf{x}_*}{\arg\min} \Phi(\mathbf{x}_*) + \frac{\alpha}{2}\|\mathbf{x}_* - \mathbf{x}_K^{i+1}\|^2.$$

17:     $i \leftarrow i + 1$
18: **end while**
19: **Output:** Reconstructed image $\mathbf{x}_*^i$ and trajectory $X^i$.

---

### 3.3. Complexity of MS-Flow

We compare the time and space complexity of MS-Flow with the single-shooting baseline D-Flow, focusing on the trajectory-consistency subproblem in Equation (10a). The data-consistency update in Equation (10b) is shared across methods. In particular, its cost depends on $\Phi$; for quadratic $\Phi(\mathbf{x}) = \frac{1}{2}\|\mathbf{A}\mathbf{x} - \mathbf{y}\|^2$ it reduces to solving a linear system.

Let $n$ be the state dimension, and $n_t$ the number of time-discretization steps for integrating the flow ODE. MS-Flow partitions the interval into $K$ shooting segments (with $K = n_t$ for explicit Euler). We denote by $\mathsf{C}_{\text{fwd}}$ and $\mathsf{C}_{\text{vjp}}$ the cost of one forward and one VJP through $v_\theta$, and by $\mathsf{M}_{\text{net}}$ the network's peak activation memory. See Appendix C for time and memory complexity calculation details.

**Time complexity.** As summarized in Table 1, D-Flow costs $O(n_t(\mathsf{C}_{\text{fwd}} + \mathsf{C}_{\text{vjp}}))$ per iteration, due to forward integration followed by backpropagation through the entire discretized trajectory. In contrast, one outer iteration of MS-Flow with $L$ backward sweep steps costs $O(LK(\mathsf{C}_{\text{fwd}} + \mathsf{C}_{\text{vjp}}))$, scaling linearly in the number of shooting points while avoiding global backpropagation through time. The Jacobian-free variant further reduces the cost to $O(LK\,\mathsf{C}_{\text{fwd}})$.

*Table 1.* Time and memory complexity comparison between D-Flow and MS-Flow. Here $n$ is the state dimension, $n_t$ the number of ODE discretization steps, $K$ the number of shooting intervals (with $K = n_t$ for explicit Euler), $L$ the number of inner iterations, $C_{\text{fwd}}$ and $C_{\text{vjp}}$ the costs of one forward and backward pass through $v_\theta$, and $M_{\text{net}}$ the peak activation memory of the network.

| Method | Time per iteration | Peak memory |
|---|---|---|
| D-Flow (single shooting) | $O(n_t(C_{\text{fwd}} + C_{\text{vjp}}))$ | $O(n_t M_{\text{net}})$ |
| D-Flow (adjoint) | $O(n_t(C_{\text{fwd}} + C_{\text{vjp}}))$ | $O(M_{\text{net}})$ |
| MS-Flow (exact gradients) | $O(LK(C_{\text{fwd}} + C_{\text{vjp}}))$ | $O(M_{\text{net}})$ |
| MS-Flow (Jacobian-free) | $O(LK\,C_{\text{fwd}})$ | $O(M_{\text{net}})$ |

**Memory complexity.** The cost for D-Flow arises from storing intermediate activations across $n_t$ solver steps, leading to peak memory $O(n_t M_{\text{net}})$, unless adjoint methods are used (Chen et al., 2018). MS-Flow introduces state variables $\{\mathbf{x}_k\}_{k=0}^{K}$, but does not require storing the full computational graph. Each shooting interval is processed locally, yielding peak memory $O(M_{\text{net}}) + O(Kn)$, which is constant w.r.t. the discretization when network activations dominate. This is evaluated in Figure 1. For example, the activations of the model in Section 5.2 require about 550MB, whereas one state $\mathbf{x}_k$ only requires 0.19MB of GPU memory.

**Parallelization.** Because the trajectory continuity is enforced locally, MS-Flow admits parallel computation across shooting intervals $F_{k,k+1}$. Forward evaluations of $v_\theta$ can be batched to trade memory for computation time. In contrast, single-shooting methods are inherently sequential in time and do not admit such parallelism. We demonstrate the parallelization of MS-Flow in Figure 3.

## 4. Properties of MS-Flow

We now establish key theoretical properties of MS-Flow. We first revisit the stability benefits of multiple shooting. Then, we prove that the inner trajectory updates in Algorithm 1 converge. This is important because MS-Flow is an inference-time optimization procedure: convergence ensures the trajectory objective decreases reliably, yielding stable and reproducible reconstructions. Finally, we show that the memory-efficient implementation based on approximate (Jacobian-free) gradients retains convergence guarantees by enforcing a sufficient-decrease condition.

### 4.1. Stability of Multiple-Shooting

A standard difficulty with single shooting is long-horizon sensitivity: if $v_\theta$ is $L$-Lipschitz in $\mathbf{x}$ (uniformly in $t$), then by Grönwall's inequality the flow map satisfies

$$\|\mathbf{x}(t; \mathbf{x}_1) - \mathbf{x}(t; \mathbf{x}_2)\|_2 \leq e^{Lt} \|\mathbf{x}_1 - \mathbf{x}_2\|_2. \quad (14)$$

Thus, perturbations can be exponentially amplified over the full time horizon (Hairer et al., 1993). Multiple shooting

alleviates this by splitting the time interval into shorter segments $0 = t_0 < \cdots < t_K = 1$ and enforcing dynamics locally. On each interval $[t_k, t_{k+1}]$ of length $\Delta_k$, the same bound applies with $t$ replaced by $\Delta_k$, yielding a local amplification factor $e^{L\Delta_k}$ (Hairer et al., 1993). In MS-Flow, this translates into better-conditioned trajectory updates: the defect penalties $\|\mathbf{x}_{k+1} - F_{k,k+1}(\mathbf{x}_k)\|_2^2$ control errors locally rather than allowing them to accumulate across the entire trajectory.

### 4.2. Convergence of Trajectory Updates

We now study the convergence properties of the coordinate descent algorithm used to solve the trajectory update in Equation (10a). At the $i$-th outer iteration of the optimization, we fix the estimate $\mathbf{x}_*^i$ and update the trajectory $X := [\mathbf{x}_0, \ldots, \mathbf{x}_K]$ by minimizing Equation (11). The backward sweep in Algorithm 2 achieves that by performing a cyclic block update in Gauss-Seidel order: for $\ell = 1, \ldots, L$, we update blocks $k = K, K-1, \ldots, 0$ as

$$\mathbf{x}_k^{(\ell)} = \mathbf{x}_k^{(\ell-1)} - \eta_k \nabla_{\mathbf{x}_k} \mathcal{J}_i(X^{(\ell,k)}). \quad (15)$$

Above $X^{(\ell,k)} = [\mathbf{x}_0^{(\ell-1)}, \ldots, \mathbf{x}_{k-1}^{(\ell-1)}, \mathbf{x}_k^{(\ell)}, \mathbf{x}_{k+1}^{(\ell)}, \ldots, \mathbf{x}_K^{(\ell)}]$ denotes the Gauss-Seidel iterate with blocks 0 to $k$ at $\ell-1$ and blocks $k+1$ to $K$ at $\ell$. That is, to update $\mathbf{x}_k^\ell$ we use the most recently updated trajectory points at later time indices $\{\mathbf{x}_{k+1}^{(\ell)}, \ldots, \mathbf{x}_K^{(\ell)}\}$. We emphasize that Equation (15) refers to the *exact* partial gradient of Equation (11). When using efficient approximations such as Jacobian-free updates, we enforce a sufficient decrease condition via a standard backtracking line search on $\mathcal{J}_i$, as discussed in Remark 4.2.

We now formalize the behavior of the backward sweep used in MS-Flow. In Proposition 4.1, we show that under regularity assumptions and appropriate step sizes, the cyclic Gauss-Seidel updates are guaranteed to decrease the trajectory objective $\mathcal{J}_i$ at every sweep. Moreover, the updates asymptotically stabilize, and any accumulation point of inner iterates is a first-order stationary point of $\mathcal{J}_i$. The proof is given in Section D.

**Proposition 4.1** (Monotone descent and stationarity with MS-Flow). *Under regularity assumption (Assumption D.1), the iterates $\{X^{(\ell)}\}_{\ell \geq 0}$ generated by Equation (15) satisfy:*

1. *Monotone decrease: $\mathcal{J}_i(X^{(\ell+1)}) \leq \mathcal{J}_i(X^{(\ell)})$ for all $\ell$.*
2. *Vanishing steps: $\sum_{\ell \geq 0} \sum_{k=0}^{K} \|\mathbf{x}_k^{(\ell+1)} - \mathbf{x}_k^{(\ell)}\|_2^2 < \infty$, hence $\|\mathbf{x}_k^{(\ell+1)} - \mathbf{x}_k^{(\ell)}\|_2 \to 0$ for every $k$.*
3. *Stationarity of limit points: every accumulation point $\bar{X}$ of $\{X^{(\ell)}\}$ is a first-order stationary point of $\mathcal{J}_i$, that is, $\nabla \mathcal{J}_i(\bar{X}) = 0$.*

*Remark* 4.2 (Inexact and Jacobian-free updates). Replacing $\nabla_{\mathbf{x}_k} \mathcal{J}_i$ in Equation (15) by an approximation removes the guarantee of descent for Equation (11). However, Proposition 4.1 still holds if $\eta_k$ is chosen via an Armijo backtracking

*Table 2.* Average computation time (seconds per image) over 100 iterations for different temporal discretizations ($n_t$), averaged over 5 images and 10 runs per image.

| Method | $n_t = 3$ | $n_t = 6$ | $n_t = 12$ |
|---|---|---|---|
| D-Flow | 7.43 | 17.06 | 175.57 |
| MS-Flow w/ JFB ($L = 1$) | 2.44 | 2.63 | 4.44 |
| MS-Flow w/ JFB ($L = 10$) | 18.65 | 22.00 | 40.10 |
| MS-Flow w/o JFB ($L = 1$) | 9.30 | 19.61 | 41.05 |
| MS-Flow w/o JFB ($L = 10$) | 86.65 | 190.70 | 405.47 |
| MS-Flow w/ GD ($L = 1$) | 4.35 | 5.36 | 9.42 |
| MS-Flow w/ GD ($L = 10$) | 38.64 | 48.23 | 89.09 |

line search that enforces a uniform sufficient decrease in the *true* objective $\mathcal{J}_i$ (Nocedal & Wright, 2006; Wright et al., 1999). More generally, these inexact Gauss-Seidel steps can be interpreted as block updates on a local upper model (first-order surrogate plus a quadratic term), similar to majorization and proximal methods, for which standard sufficient-decrease arguments imply that bounded iterates converge to stationary points (Parikh et al., 2014; Nocedal & Wright, 2006; Wright et al., 1999). We empirically validate this in Figure 3.

# 5. Experiments

In this section, we evaluate the performance of our MS-Flow across a wide range of applications, including computed tomography, deblurring, inpainting, and super-resolution. We consider various baselines described in each experiment. Our goal is to address the following questions:

**(Q1)** How stable is MS-Flow, and how does it compare with existing techniques?

**(Q2)** How efficient is MS-Flow?

**(Q3)** What is the downstream performance of MS-Flow compared with leading methods for solving inverse problems with flow-based models?

Throughout all experiments, we optimize the objective defined in Equation (7). We provide additional information on our experimental settings in Section E. The code is publicly available at: https://github.com/alexdenker/MS-Flow

## 5.1. Empirical Convergence and Efficiency Analysis

We first validate the convergence properties of our optimization scheme and demonstrate the robustness and computational efficiency of MS-Flow.

**Setup.** We consider sparse-angle tomography on the OrganCMNIST dataset (Yang et al., 2021; 2023) to study algorithmic design choices and to compare with D-Flow. The forward operator is the Radon transform with 18 equidistant

angles across $[0, \pi)$, adding random Gaussian noise with a $\sigma_{\text{noise}} = 0.01$. The training is implemented using the Flow Matching library (Lipman et al., 2023). We use this dataset to study both **(Q1)** and **(Q2)** and provide ablations.

**Empirical Convergence.** We provide an empirical validation for the convergence result in Proposition 4.1 in Figure 3. In particular, we compare three methods for optimizing the trajectory term in Equation (11): (i) Coordinate descent with the Jacobian-free approximation (CD with JFB); (ii) Coordinate descent with exact gradients (CD without JFB); and (iii) Gradient descent. We note that, in early iterations, the convergence is similar and only in later iterations we observe an advantage (albeit minor) of using exact gradients. However, performance is dominated by the first few updates, where all three methods behave similarly. In all experiments, we use $L \leq 10$, and we adopt CD with JFB as the default inner-loop optimizer for the remainder of the paper. This choice is supported by both theory and practice: in many alternating optimization settings, subproblems do not need to be solved exactly to ensure overall convergence, and a small number of gradient steps per subproblem is often sufficient to make consistent progress, and is computationally efficient (Eckstein & Bertsekas, 1992; Boyd et al., 2011). We can show convergence for CD with JFB when a line search is employed for the step size, see Remark 4.2. Empirically, we observe that a constant step size suffices to obtain stability.

**Scaling and Robustness Analysis.** We also study the convergence for various choices of the number of discretization points $n_t$ (equal to the number of shooting points $K$) and the choice of the regularization strength $\lambda$ for the radial Gaussian prior in Equation (8). The results are reported in Figure 2, showing that MS-Flow is stable for a wide range of hyperparameters. While D-Flow can achieve a similar peak PSNR, we often see a performance deterioration over the iterations. We further compare the computational time in Table 2 for different choices of $n_t$ and different numbers of inner coordinate descent iterations for MS-Flow ($L = 1$ and $L = 10$). We see that because of the end-to-end nature of D-Flow, its computational time increases drastically with the number of discretization points $n_t$. In contrast, our MS-Flow with the Jacobian-free approximation is the fastest method and scales gracefully with the number of discretization points $n_t$.

## 5.2. Image Recovery Tasks

Next, we demonstrate that MS-Flow is competitive with other recently proposed flow-based inverse problem solvers.

**Setup.** In this set of experiments, we consider three common image recovery tasks on the CelebA dataset (Yang et al., 2015). Specifically, we evaluate on: (i) image deblurring with a Gaussian blur with intensity $\sigma = 1$ and kernel size 61; (ii) super-resolution with bicubic interpolation with scale

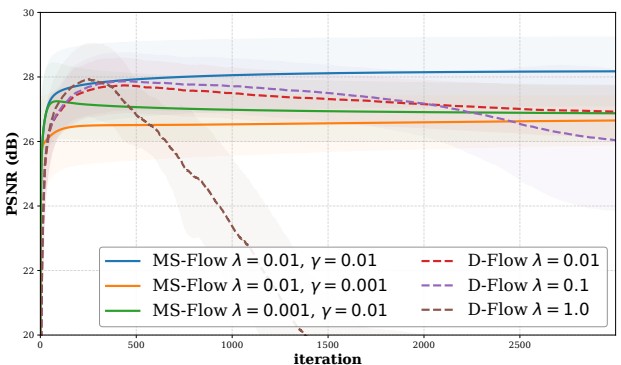
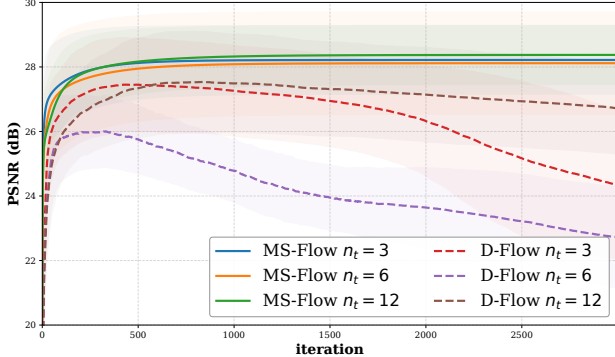

*(a)* Changing the regularization terms.

*(b)* Changing the time discretization $n_t$.

*Figure 2.* Evaluation on the OrganCMNIST for sparse-angle CT. Left: The effect of regularization terms. Right: Comparison of MS-Flow with D-Flow across temporal resolutions. Shaded areas represent the standard deviation over the 10 images.

*Table 3.* Quantitative comparison of different reconstruction methods across three image recovery tasks. Best results per task and metric (PSNR, SSIM, LPIPS) are highlighted in **bold**, and second-best are underlined.

| | Gaussian Deblurring | | | Box-Inpainting | | | 2x Super-Resolution | | | 4x Super-Resolution | | | **Time/img** |
|---|---|---|---|---|---|---|---|---|---|---|---|---|---|
| | PSNR (↑) | SSIM (↑) | LPIPS (↓) | PSNR (↑) | SSIM (↑) | LPIPS (↓) | PSNR (↑) | SSIM (↑) | LPIPS (↓) | PSNR (↑) | SSIM (↑) | LPIPS (↓) | [s] |
| Flow-Prior | 36.28 | 0.965 | **0.009** | 35.48 | 0.981 | **0.011** | 34.34 | 0.948 | 0.015 | 27.27 | 0.796 | 0.094 | 24.30 |
| OT-ODE | 37.72 | **0.971** | 0.015 | 33.44 | 0.972 | 0.013 | 35.14 | **0.960** | **0.008** | 28.18 | 0.834 | 0.052 | 5.19 |
| PnP-Flow | 36.24 | 0.965 | 0.049 | **35.87** | **0.983** | **0.011** | 34.64 | 0.950 | 0.059 | 28.28 | 0.840 | 0.168 | 5.97 |
| D-Flow | 36.07 | 0.957 | 0.029 | 34.20 | 0.967 | 0.012 | **35.49** | 0.951 | 0.019 | 29.93 | 0.873 | 0.062 | 26.31 |
| MS-Flow (Ours) | **39.04** | 0.968 | **0.009** | 34.77 | 0.974 | 0.015 | **36.41** | 0.957 | 0.023 | 28.52 | 0.838 | 0.091 | 14.11 |

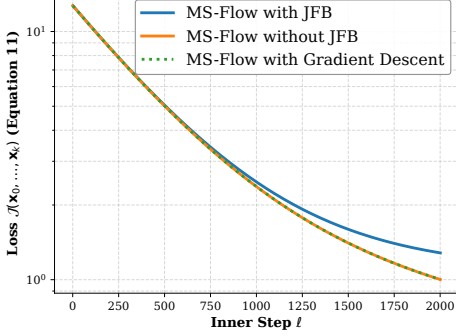

*Figure 3.* Convergence of the trajectory loss in Equation (11) for coordinate descent (with and without the Jacobian-free approximation) and full gradient descent.

factor 2; and (iii) inpainting with a central box mask of size $32 \times 32$ pixels. In all cases we add Gaussian noise with $\sigma_{\text{noise}} = 0.01$. We use standard imaging metrics, namely PSNR and SSIM (Wang et al., 2004), and report the values obtained for the reconstruction over 50 images from CelebA. This setting both addresses **(Q3)** as well as provides a direct comparison with several flow-based inverse problem solvers, including Flow-Prior (Zhang et al., 2024), OT-ODE (Pokle et al., 2024), PnP-Flow (Martin et al., 2025), and D-Flow.

**Results.** Experimental results are summarized in Table 3 and reconstructions are provided in Figure 4. For both Gaussian deblurring and super-resolution, MS-Flow achieves the highest PSNR, with a small drop in SSIM. This can

be explained by our data-consistency step in (10b), which includes a regularizer penalizing deviations from the terminal shooting point $\mathbf{x}_K$. The deviation is measured with the Euclidean norm, which also promotes smoothness, resulting in a higher (better) PSNR. For Gaussian deblurring and super-resolution, we show the initialization of the shooting points and the final converged trajectory in Figures 7 and 8.

## 6. Latent Flow Models

We now demonstrate that in addition to standard flow-matching networks, our MS-Flow scales to modern large-scale latent flow models by evaluating it on Stable Diffusion 3.5 (Esser et al., 2024).

**Setup.** We compare our MS-Flow with FlowDPS (Kim et al., 2025) and FLAIR (Erbach et al., 2025), two flow-based inverse problem solvers that avoid backpropagation through the flow and therefore scales to large latent flow architectures. For MS-Flow, we again consider the Jacobian-free gradient approximation for the trajectory update. We consider Gaussian deblurring with blur standard deviation $\sigma = 3.0$.

In latent flow models, the flow operates in latent space while observations are defined in pixel space. Let $D : \mathcal{Z} \rightarrow \mathcal{X}$ denote the decoder mapping latent variables to images, with $\mathcal{Z}$ as the latent space and $\mathcal{X}$ as the pixel space. In this setting

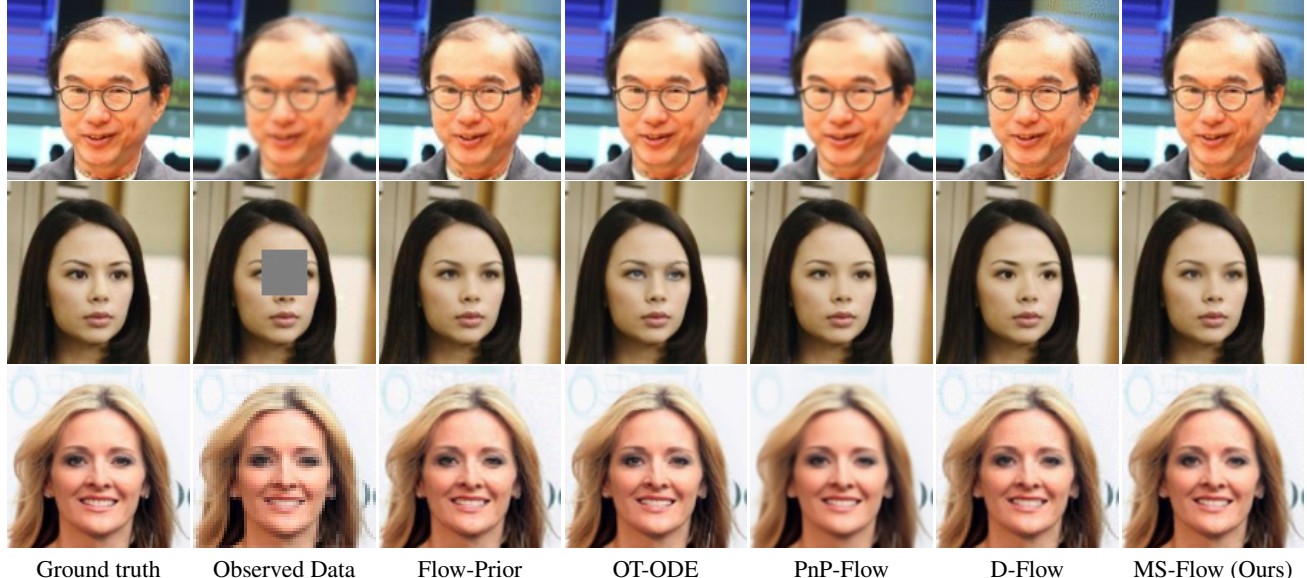

| Ground truth | Observed Data | Flow-Prior | OT-ODE | PnP-Flow | D-Flow | MS-Flow (Ours) |

*Figure 4.* Comparison of the reconstructions for the three image recovery tasks on CelebA. First row: Gaussian deblurring. Second row: Inpainting. Third row: Super-resolution.

the data-consistency update from Equation (10b) becomes

$$\mathbf{z}_*^{i+1} = \arg\min_{\mathbf{z}_*} \frac{1}{2}\|\mathbf{A}D(\mathbf{z}_*) - \mathbf{y}\|_2^2 + \frac{\alpha}{2}\|\mathbf{z} - \mathbf{z}_K^{i+1}\|_2^2.$$

All optimization variables are defined in latent space and are denoted by $\mathbf{z}$. The final image reconstruction is obtained as $\mathbf{x}_* = D(\mathbf{z}_*)$. As the decoder $D$ is nonlinear, the objective no longer admits a closed-form solution. We therefore solve it using a small number of gradient descent steps with Adam (Kingma & Ba, 2015). We refer to Appendix F for details.

**Results.** Our results are reported in Table 4 for different noise levels, $\sigma_{\text{noise}} = 0.01$ and $\sigma_{\text{noise}} = 0.1$. We report both the mean PSNR and mean SSIM, averaged over the first 100 images of the FFHQ dataset (Karras et al., 2019). Across both settings, MS-Flow consistently outperforms FlowDPS in terms of PSNR, SSIM and LPIPS and achieves comparable results to FLAIR. For $\sigma_{\text{noise}}$, we observe a distortion–perception trade-off: MS-Flow yields lower distortion, as reflected by higher PSNR and SSIM, whereas FLAIR provides better perceptual quality, indicated by lower LPIPS. Qualitative reconstructions are shown in Figure 5. These results further show the contribution of MS-Flow as an efficient and effective approach for utilizing flow-based models for solving inverse problems.

## 7. Discussion and Conclusions

We introduce MS-Flow, a multiple-shooting framework for LSO with flow-based models. By representing the trajectory as a sequence of intermediate shooting states and enforcing the flow dynamics locally via trajectory-matching

*Table 4.* Comparison for Gaussian deblurring on FFHQ for different noise levels. Best results in bold, and second-best are underlined.

| Method | $\sigma_{\text{noise}} = 0.01$ | | | $\sigma_{\text{noise}} = 0.1$ | | |
| | PSNR (↑) | SSIM (↑) | LPIPS (↓) | PSNR (↑) | SSIM (↑) | LPIPS (↓) |
|---|---|---|---|---|---|---|
| Degraded Image | 26.32 | 0.763 | 0.388 | 23.20 | 0.335 | 0.684 |
| FlowDPS | 27.78 | 0.751 | 0.349 | 26.87 | 0.697 | 0.366 |
| FLAIR | 29.18 | 0.783 | **0.199** | **28.93** | **0.775** | **0.200** |
| MS-Flow (Ours) | **30.65** | **0.837** | 0.268 | 28.62 | 0.772 | 0.403 |

constraints, MS-Flow overcomes two key limitations of single-shooting approaches: high memory consumption and poor conditioning. Empirically, MS-Flow achieves competitive or improved reconstruction performance across a range of imaging inverse problems. It maintains constant memory usage regardless of the number of ODE discretization steps, making it significantly more efficient than common existing techniques. This positions MS-Flow as a well-suited approach for high-resolution problems and fine-grained temporal discretizations, such as image restoration for FFHQ with Stable Diffusion 3.5, where single-shooting approaches become impractical.

Relaxing the global ODE constraint in MS-Flow naturally introduces a small set of design choices, most notably the penalty weights and the number of shooting points. In practice, we found that simple fixed penalties work reliably across tasks, indicating that MS-Flow is robust to these settings. Moreover, for inverse problems, which are at the focus of this work, the intermediate shooting states act as auxiliary variables, so performance is governed primarily by the terminal state rather than full-trajectory fidelity. Building on this, incorporating adaptive penalty updates, as in

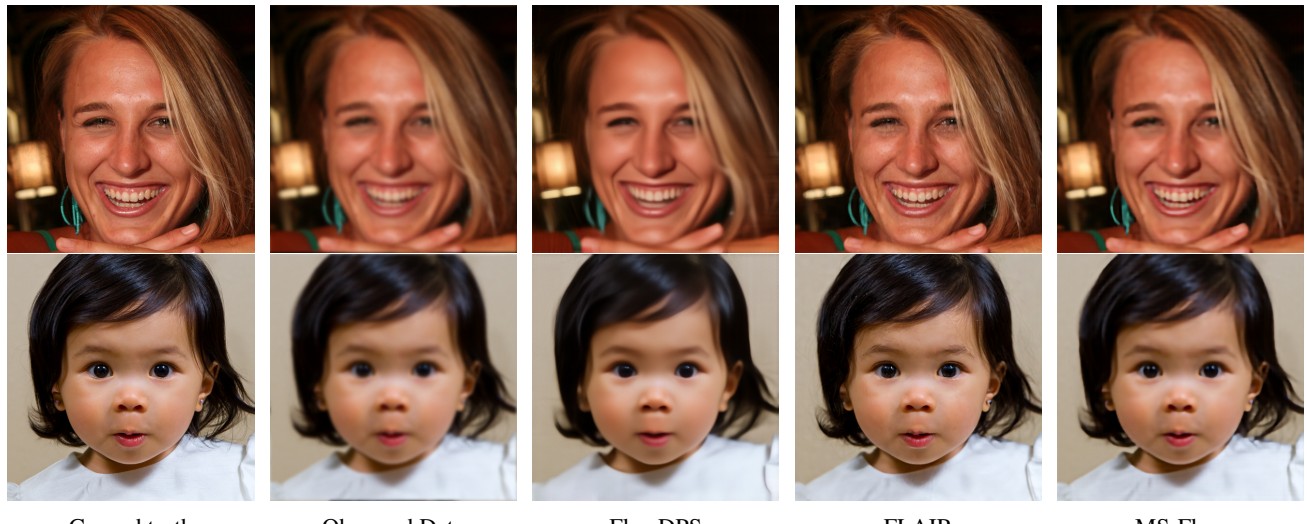

Ground truth  Observed Data  FlowDPS  FLAIR  MS-Flow

*Figure 5.* Reconstruction for Gaussian deblurring on FFHQ with $\sigma_{\text{noise}} = 0.01$.

splitting-based optimization methods (Boyd et al., 2011), in this context, is an interesting future work direction.

A promising next step is to extend MS-Flow to SDE-based generative models, including diffusion and score-based methods (Song et al., 2021b). By introducing appropriate stochastic consistency constraints across segments, MS-Flow could provide a practical route to memory-efficient conditional sampling in these models.

## Acknowledgments

AD, CBS acknowledges support from the EPSRC (EP/V026259/1). AD additionally acknowledges support from DESY (Hamburg, Germany), a member of the Helmholtz Association HGF. ZK acknowledges support from the EPSRC (EP/X010740/1).

## Impact Statement

This work aims to enhance controllability in flow-based generative models. While improved control could potentially be misused to produce misleading or harmful content, including deepfakes, the method does not expand the underlying generative capacity beyond existing models.

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

# A. Related Work

**Latent-Space Optimization with Diffusion Models** Recent work has made use of diffusion models for LSO to solve inverse problems. BIRD (Chihaoui et al., 2024) applies this concept for blind image restoration and jointly estimate the reconstruction and the parameters of the (unknown) degradation operator. They use the deterministic DDIM (Song et al., 2021a) sampling scheme and optimize the initial noise input. They directly backpropagate the gradients through the DDIM sampling scheme and make use of 10 sampling steps in most of their experiments. DMPlug (Wang et al., 2024a) proposes a similar framework, i.e., optimizing the initial latent code such that the output of the deterministic DDIM sampler is consistent to measurements. In their experiments the authors make use of 3 DDIM sampling steps to parametrise the reverse process. The memory requirements of both approaches scales like D-Flow (Ben-Hamu et al., 2024), see Section 3.3.

**Flow-based Solvers for Inverse Problems.** A growing line of work uses *pre-trained* flow-matching and rectified-flow models as implicit priors for inverse problems by modifying the sampling dynamics to incorporate data consistency, rather than retraining the model. In the setting of linear inverse problems, Pokle et al. (2024) propose a training-free solver for inverse problems using pre-trained flow models, with theoretically motivated weighting and conditional optimal transport (OT) paths. More recently, guidance schemes specialized to flow matching have appeared: Yan et al. (2025) introduce interpolant guidance (FIG) that steers reverse-time sampling using measurement interpolants; and Kim et al. (2025) adapt posterior-sampling ideas to flows by decomposing the flow ODE into clean/noise components (via a flow analogue of Tweedie-based relations) and injecting likelihood gradients and stochasticity to sample from the posterior (Chung et al., 2023). Complementary to guidance-based posterior sampling, plug-and-play hybrids have also been explored: Martin et al. (2025) defines a time-dependent denoiser induced by a pre-trained flow-matching model and alternates data-fidelity updates with projections/denoising along the learned path. Finally, conditional and invertible modeling has a longer history in inverse problems: Ardizzone et al. (2019) uses conditional invertible neural networks to represent inverse maps and uncertainty, illustrating the tradeoff between task-specific conditional training and test-time inference with a fixed unconditional prior.

**Guidance and Optimal Control for Generative Flows.** Beyond inverse problems, several works formulate *controlled generation* for ODE-based generators as test-time trajectory optimization, often by differentiating through the sampler. Liu et al. (2023) shows how to backpropagate guidance gradients to intermediate times of a generative ODE, enabling controllable generation without retraining, while Ben-Hamu et al. (2024) (D-Flow) differentiates through the flow sampler to optimize control objectives over the generated sample. Patel et al. (2025) propose a gradient-free steering and inversion strategy for rectified-flow models and demonstrate applications including linear inverse problems and editing. From an optimal-control perspective, Wang et al. (2024b) presents a training-free guided flow-matching framework derived from optimal control, interprets backprop-through-ODE guidance as special cases, and provides algorithms for guided generation.

**Multiple-Shooting/Single-Shooting in ML.** Single-shooting is the default in neural ODEs and generative flows: one optimizes parameters (and possibly an initial latent) while enforcing dynamics by a single forward solve over the full time horizon (Chen et al., 2018). This can become numerically unstable for long horizons, stiff dynamics, or when strong guidance terms create sharp transients. Multiple-shooting, a classical remedy from optimal control (Bock & Plitt, 1984), introduces intermediate states as optimization variables and enforces continuity constraints between segments, improving stability and enabling time-parallelism. In modern ML, Massaroli et al. (2021) formalize differentiable multiple-shooting layers as implicit models with parallelizable root finding, and Turan & Jäschke (2021) study multiple-shooting training for neural ODEs on time-series benchmarks. Our approach aligns with this viewpoint by splitting the generative trajectory into segments and enforcing (soft) continuity, trading a larger but better-conditioned optimization for improved robustness. The recently proposed DRIP framework (Eliasof et al., 2023) also implements a similar backward sweep to our Algorithm 2. However, we use a fixed pre-trained flow model, whereas DRIP also learns the underlying dynamics.

**Intermediate Layer Optimization (ILO).** Prior work has explored intermediate-state optimization as an alternative to end-to-end backpropagation through deep networks. Daras et al. (Daras et al., 2021) introduce a layer-wise optimization scheme. Rather than propagating gradients through the entire network, optimization proceeds sequentially over intermediate layers (or states), starting from the output layer and moving backward toward the input. This idea has recently been extended to diffusion models in DMILO (Zheng et al., 2025), where intermediate DDIM states are optimized in a sequential manner. ILO has notable differences to our multiple-shooting variant. In particular, DMILO requires sequential optimization of intermediate DDIM satates, whereas our MS-Flow jointly optimizes the full trajectory. Moreover, our JFB variant eliminates the need for backpropagation through diffusion steps entirely, yielding a fully backpropagation-free optimization procedure.

## B. Numerical Details

### B.1. Explicit Euler Discretization

In Algorithm 2 we used the notation $F_{k,k+1}(\mathbf{x}_k)$ for the integration of the dynamics from $t_k$ to $t_{k+1}$ starting at $\mathbf{x}_k$. We can realize this using a single explicit Euler step with $\Delta_k = t_{k+1} - t_k$:

$$F_{k+1,k}^{\text{EE}}(\mathbf{x}) := \mathbf{x} + \Delta_k v_\theta(\mathbf{x}, t_k). \tag{16}$$

Hence, in terms of number of function evaluations (NFEs), each step requires one NFE, different than the typical midpoint discretization used in D-Flow, that requires two NFEs per step. Substituting this into the trajectory loss in Equation (11), we obtain

$$\mathcal{J}^{\text{EE}}(\mathbf{x}_0, \ldots, \mathbf{x}_K) := \frac{\alpha}{2}\|\mathbf{x}_*^i - \mathbf{x}_K\|^2 + \lambda \mathcal{R}(\mathbf{x}_0) + \frac{\gamma}{2}\sum_{k=1}^{K}\|\mathbf{x}_k - [\mathbf{x}_{k-1} + \Delta_{k-1}v_\theta(\mathbf{x}_{k-1}, t_{k-1})]\|^2. \tag{17}$$

Following the coordinate-wise backward sweep introduced in Section 3.2, we have to solve the following optimization problem

$$\tilde{\mathbf{x}}_k = \arg\min_u \mathcal{J}_k(u) := \frac{\gamma}{2}\|u - z_k\|_2^2 + \frac{\gamma}{2}\|\tilde{\mathbf{x}}_{k+1} - (u + \Delta_k v_\theta(u, t_k))\|^2 \tag{18}$$

for $k = K-1, \ldots, 1$ and $z_k = F_{k-1,k}^{\text{EE}}(\mathbf{x}_{k-1})$. The gradient is given by

$$\nabla_u \mathcal{J}_k(u) = (u - z_k) - (I + \Delta_k J_{v_\theta}(u))^T(\tilde{\mathbf{x}}_{k+1} - (u + \Delta_k v_\theta(u, t_k))), \tag{19}$$

with $\tilde{\mathbf{x}}_{k+1}$ being the minimizer for $\mathcal{J}_{k+1}$. Computing this gradient directly requires one backward pass of the flow model. However, in our experiments, we observe that we can approximate $I + \Delta t_k J_{v_\theta} \approx I$. In diffusion models, neglecting the Jacobian of the model is quite standard, see e.g. (Poole et al., 2023). Note that this approximation also gets better if $\Delta t_k \approx 0$, i.e., if we increase the number of shooting points. Using this approximation, we obtain a Jacobian-free gradient surrogate

$$\nabla_u \mathcal{J}_k(u) \approx (u - z_k) - (\tilde{\mathbf{x}}_{k+1} - (u + \Delta_k v_\theta(u, t_k))),$$

which numerically still gives stable convergence.

### B.2. Trajectory Optimization

We compute the gradient for all intermediate control points as:

$$\nabla_{\mathbf{x}_K}\mathcal{J} = -\alpha(\mathbf{x}_*^i - \mathbf{x}_K) + \gamma(\mathbf{x}_K - F_{k-1,k}(\mathbf{x}_{k-1})), \tag{20}$$

$$\begin{aligned}\nabla_{\mathbf{x}_k}\mathcal{J} = &-\gamma J_{F_{k,k+1}}(\mathbf{x}_k)^T(\mathbf{x}_{k+1} - F_{k,k+1}(\mathbf{x}_k)) \\ &+ \gamma(\mathbf{x}_k - F_{k-1,k}(\mathbf{x}_{k-1})), \ k = 1, \ldots, K-1,\end{aligned} \tag{21}$$

$$\nabla_{\mathbf{x}_0}\mathcal{J} = -\gamma J_{F_{0,1}}(\mathbf{x}_0)^T(\mathbf{x}_1 - F_{0,1}(\mathbf{x}_0)) + \gamma \nabla \mathcal{R}(\mathbf{x}_0) \tag{22}$$

Here $J_{F_{k,k+1}}$ denotes the Jacobian of the flow update from $t_k$ to $t_{k+1}$. Given the trajectory $\{\mathbf{x}_0^{(\ell-1)}, \ldots, \mathbf{x}_K^{(\ell-1)}\}$ and step size $\eta$.

1. **Update the Final Shooting Point:** The gradient for $\mathbf{x}_K$ is:

   $$\nabla_{\mathbf{x}_K}\mathcal{J} = -\alpha(\mathbf{x}_*^i - \mathbf{x}_K) + \gamma(\mathbf{x}_K - F_{K-1,K}(\mathbf{x}_{K-1}^{(\ell-1)}))$$

   The update rule is:

   $$\mathbf{x}_K^{(\ell)} \leftarrow \mathbf{x}_K^{(\ell-1)} - \eta \cdot \nabla_{\mathbf{x}_K}\mathcal{J}(\mathbf{x}_K^{(\ell-1)}, \mathbf{x}_{K-1}^{(\ell-1)}) \tag{23}$$

2. **Backward Sweep for Intermediate Shooting Points:** We iterate backward, using the newly computed neighbour $\mathbf{x}_{k+1}^{(\ell)}$ in the gradient calculation. For $k = K-1, K-2, \ldots, 1$:

   $$\nabla_{\mathbf{x}_k}\mathcal{J} = -\gamma J_{F_{k,k+1}}(\mathbf{x}_k)^T(\mathbf{x}_{k+1}^{(\ell)} - F_{k,k+1}(\mathbf{x}_k)) + \gamma(\mathbf{x}_k - F_{k-1,k}(\mathbf{x}_{k-1}^{(\ell-1)}))$$

   The update rule is:

   $$\mathbf{x}_k^{(\ell)} \leftarrow \mathbf{x}_k^{(\ell-1)} - \eta \cdot \nabla_{\mathbf{x}_k}\mathcal{J}(\mathbf{x}_k^{(\ell-1)}, \mathbf{x}_{k-1}^{(\ell-1)}, \mathbf{x}_{k+1}^{(\ell)}) \tag{24}$$

---

**Algorithm 2** A single iteration of coordinate descent for trajectory optimization

---

1: **Input:** Initial trajectory $(\mathbf{x}_0^{\text{old}}, \ldots, \mathbf{x}_K^{\text{old}})$, target state $\mathbf{x}_*^i$, dynamics $F_{k-1,k}$, weights $\alpha, \lambda, \gamma$, step size $\eta$.
2: {*Forward Sweep:* Compute consistency targets $\mathbf{z}_k$ using old trajectory.}
3: **for** $k = 1$ **to** $K$ **do**
4:     $\mathbf{z}_k \leftarrow F_{k-1,k}(\mathbf{x}_{k-1}^{\text{old}})$
5: **end for**
6: {*Backward Sweep:* Update control points from $K$ down to $0$ using updated neighbors.}
7: **for** $k = K$ **down to** $0$ **do**
8:     **if** $k = K$ **then**
9:         Compute $\nabla_{\mathbf{x}_K} \mathcal{J}$ using $\mathbf{z}_K$:
10:            $\nabla_{\mathbf{x}_K} \mathcal{J} = -\alpha(\mathbf{x}_*^i - \mathbf{x}_K^{\text{old}}) + \gamma(\mathbf{x}_K^{\text{old}} - \mathbf{z}_K)$
11:        $\mathbf{x}_K^{\text{new}} \leftarrow \mathbf{x}_K^{\text{old}} - \eta \nabla_{\mathbf{x}_K} \mathcal{J}$
12:    **else if** $0 < k < K$ **then**
13:        Compute $\nabla_{\mathbf{x}_k} \mathcal{J}$ using $\mathbf{x}_{k+1}^{\text{new}}$ and $\mathbf{z}_k$:
14:            $\nabla_{\mathbf{x}_k} \mathcal{J} = \gamma(\mathbf{x}_k^{\text{old}} - \mathbf{z}_k) - \gamma J_{F_{k,k+1}}(\mathbf{x}_k^{\text{old}})^T(\mathbf{x}_{k+1}^{\text{new}} - F_{k,k+1}(\mathbf{x}_k^{\text{old}}))$
15:        $\mathbf{x}_k^{\text{new}} \leftarrow \mathbf{x}_k^{\text{old}} - \eta \nabla_{\mathbf{x}_k} \mathcal{J}$
16:    **else if** $k = 0$ **then**
17:        Compute $\nabla_{\mathbf{x}_0} \mathcal{J}$ using $\mathbf{x}_1^{\text{new}}$:
18:            $\nabla_{\mathbf{x}_0} \mathcal{J} = -\gamma J_{0,1}(\mathbf{x}_0^{\text{old}})^T(\mathbf{x}_1^{\text{new}} - F_{0,1}(\mathbf{x}_0^{\text{old}})) + \lambda \nabla \mathcal{R}(\mathbf{x}_0^{\text{old}})$
19:        $\mathbf{x}_0^{\text{new}} \leftarrow \mathbf{x}_0^{\text{old}} - \eta \nabla_{\mathbf{x}_0} \mathcal{J}$
20:    **end if**
21:    $\mathbf{x}_k^{\text{old}} \leftarrow \mathbf{x}_k^{\text{new}}$ {Store new value for subsequent steps}
22: **end for**
23: **Output:** Updated trajectory $(\mathbf{x}_0^{\text{new}}, \ldots, \mathbf{x}_K^{\text{new}})$.

---

3. **Update the Initial Shooting Point:** The update for $\mathbf{x}_0$ uses the newly computed $\mathbf{x}_1^{(\ell)}$ and includes the regularization gradient $\nabla \mathcal{R}(\mathbf{x}_0)$.

$$\nabla_{\mathbf{x}_0} \mathcal{J} = -\gamma J_{F_{0,1}}(\mathbf{x}_0)^T(\mathbf{x}_1^{(\ell)} - F_{0,1}(\mathbf{x}_0)) + \lambda \nabla \mathcal{R}(\mathbf{x}_0)$$

The update rule is:

$$\mathbf{x}_0^{(\ell)} \leftarrow \mathbf{x}_0^{(\ell-1)} - \eta \cdot \nabla_{\mathbf{x}_0} \mathcal{J}(\mathbf{x}_0^{(\ell-1)}, \mathbf{x}_1^{(\ell)}) \tag{25}$$

The trajectory for the next iteration is then $\{\mathbf{x}_0^{(\ell)}, \ldots, \mathbf{x}_K^{(\ell)}\}$. This process is repeated until convergence. The algorithm is also described in Algorithm 2.

## C. Detailed Complexity Analysis of MS-Flow

This appendix provides a detailed breakdown of time and memory for the summary in Section 3.3.

### C.1. Single-Shooting D-Flow

A gradient update in D-Flow optimizes the initial state $\mathbf{x}_0$ of the ODE-constrained problem in Equation (6). This requires:

  (i)  forward integration of the ODE for $n_t$ discretization steps, and
  (ii) differentiation through the entire solver.

With standard reverse-mode differentiation, the resulting cost per iteration is $O(n_t(\mathsf{C}_{\text{fwd}} + \mathsf{C}_{\text{vjp}}))$, while peak memory scales as $O(n_t \mathsf{M}_{\text{net}})$, due to storing intermediate states and network activations. Adjoint methods reduce memory to $O(\mathsf{M}_{\text{net}})$ but introduce additional backward ODE solves and increased wall-clock time.

### C.2. Multiple-Shooting MS-Flow

We consider one outer iteration of Algorithm 1, focusing on the trajectory update Equation (10a). Using $L$ inner coordinate-descent sweeps, each sweep consists of:

(i) a forward sweep computing $F_{k-1,k}(\mathbf{x}_{k-1})$ for $k = 1, \dots, K$, and

(ii) a backward sweep updating the block variables $\{\mathbf{x}_k\}$ using only local neighbors.

For explicit Euler with $\Delta_k = t_{k+1} - t_k$ the transition map $F_{k,k+1}$ is given as

$$F_{k,k+1}(\mathbf{x}) = \mathbf{x} + \Delta_k\, v_\theta(\mathbf{x}, t_k).$$

Evaluating all $F_{k,k+1}$ costs $O(K\, \mathsf{C}_{\mathrm{fwd}})$. Exact gradient updates additionally require computing VJPs $J_{F_{k,k+1}}(\mathbf{x}_k)^\top \mathbf{v}$. Hence, one sweep costs $O(K(\mathsf{C}_{\mathrm{fwd}} + \mathsf{C}_{\mathrm{vjp}}))$, and one outer iteration costs $O(LK(\mathsf{C}_{\mathrm{fwd}} + \mathsf{C}_{\mathrm{vjp}}))$.

**Jacobian-Free Updates**   The Jacobian for the explicit Euler is given as

$$J_{F_{k,k+1}}(\mathbf{x}) = I + \Delta_k J_{v_\theta(\mathbf{x}, t_k)}.$$

As discussed in the main text, we omit $J_{v_\theta(\mathbf{x}, t_k)}$ and use the approximation $J_{F_{k,k+1}}(\mathbf{x}) \approx I$, which eliminates the VJP computation. Under this approximation, one sweep costs $O(K\, \mathsf{C}_{\mathrm{fwd}})$.

**Memory Scaling**   MS-Flow stores the explicit trajectory variables $\{\mathbf{x}_k\}_{k=0}^K$, requiring $O(Kn)$ memory, which is negligible compared to network activations in most networks, as the number of parameters is often orders of magnitude larger than the input dimension. Crucially, gradients depend only on local segments, so the full computational graph is never materialized. The resulting peak activation memory is $O(\mathsf{M}_{\mathrm{net}})$, independent of $K$, as observed empirically in Figure 1.

## D. Assumptions and Proofs

In this section, we outline the assumptions and proofs of our results and discussions of the properties of MS-Flow in Section 4.

**Assumption D.1** (Regularity for block Gauss-Seidel descent). For the fixed outer iterate $\mathbf{x}_*^i$, assume:

1. $\mathcal{J}_i$ is bounded below on $\mathbb{R}^{n(K+1)}$.

2. $\mathcal{J}_i$ is continuously differentiable.

3. (Block Lipschitz gradients) For each $k \in \{0, \dots, K\}$ there exists $L_k > 0$ such that, for any vectors $u, v \in \mathbb{R}^n$ and any fixed values of the other blocks,

$$\left\| \nabla_{\mathbf{x}_k} \mathcal{J}_i(\dots, u, \dots) - \nabla_{\mathbf{x}_k} \mathcal{J}_i(\dots, v, \dots) \right\|_2 \le L_k \|u - v\|_2.$$

4. The step sizes satisfy $0 < \eta_k \le 1/L_k$ for all $k$ (or are chosen by backtracking line search guaranteeing sufficient decrease).

*Proof of Proposition 4.1.* For a fixed outer iterate $\mathbf{x}_*^i$, consider the inner objective $\mathcal{J}_i$ and one Gauss-Seidel sweep $\ell$. For notational convenience, define the intermediate Gauss-Seidel iterates within sweep $\ell$ by

$$X^{(\ell, K)} := X^{(\ell-1)}, \qquad X^{(\ell, -1)} := X^{(\ell)},$$

and for each $k = K, K-1, \dots, 0$ let $X^{(\ell, k)}$ be the pre-update iterate used in Equation (15),

$$X^{(\ell, k)} = [\mathbf{x}_0^{(\ell-1)}, \dots, \mathbf{x}_{k-1}^{(\ell-1)}, \mathbf{x}_k^{(\ell-1)}, \mathbf{x}_{k+1}^{(\ell)}, \dots, \mathbf{x}_K^{(\ell)}],$$

and let $X^{(\ell, k-1)}$ denote the post-update iterate obtained after updating block $k$ (so $X^{(\ell, k-1)}$ differs from $X^{(\ell, k)}$ only in block $k$).

**Step 1: Descent for a single block update.** Fix a sweep index $\ell$ and a block index $k$. Define the single-block function

$$\varphi_k(u) := \mathcal{J}_i(\mathbf{x}_0^{(\ell-1)}, \ldots, \mathbf{x}_{k-1}^{(\ell-1)}, u, \mathbf{x}_{k+1}^{(\ell)}, \ldots, \mathbf{x}_K^{(\ell)}).$$

By Assumption D.1(3), $\nabla \varphi_k$ is $L_k$-Lipschitz. The update Equation (15) is exactly a gradient step on $\varphi_k$:

$$\mathbf{x}_k^{(\ell)} = \mathbf{x}_k^{(\ell-1)} - \eta_k \nabla \varphi_k(\mathbf{x}_k^{(\ell-1)}), \qquad 0 < \eta_k \leq \frac{1}{L_k}.$$

By the descent lemma for $L_k$-smooth functions, for any $y$,

$$\varphi_k(y) \leq \varphi_k(x) + \langle \nabla \varphi_k(x), y - x \rangle + \frac{L_k}{2} \|y - x\|_2^2.$$

Applying this with $x = \mathbf{x}_k^{(\ell-1)}$ and $y = \mathbf{x}_k^{(\ell)} = x - \eta_k \nabla \varphi_k(x)$ yields

$$\varphi_k(\mathbf{x}_k^{(\ell)}) \leq \varphi_k(\mathbf{x}_k^{(\ell-1)}) - \eta_k \|\nabla \varphi_k(\mathbf{x}_k^{(\ell-1)})\|_2^2 + \frac{L_k \eta_k^2}{2} \|\nabla \varphi_k(\mathbf{x}_k^{(\ell-1)})\|_2^2$$

$$= \varphi_k(\mathbf{x}_k^{(\ell-1)}) - \left(\eta_k - \frac{L_k \eta_k^2}{2}\right) \|\nabla \varphi_k(\mathbf{x}_k^{(\ell-1)})\|_2^2$$

$$= \varphi_k(\mathbf{x}_k^{(\ell-1)}) - \left(\frac{1}{\eta_k} - \frac{L_k}{2}\right) \|\mathbf{x}_k^{(\ell)} - \mathbf{x}_k^{(\ell-1)}\|_2^2,$$

where in the last line we used $\mathbf{x}_k^{(\ell)} - \mathbf{x}_k^{(\ell-1)} = -\eta_k \nabla \varphi_k(\mathbf{x}_k^{(\ell-1)})$. Translating back to $\mathcal{J}_i$, this is

$$\mathcal{J}_i\big(X^{(\ell,k-1)}\big) \leq \mathcal{J}_i\big(X^{(\ell,k)}\big) - c_k \|\mathbf{x}_k^{(\ell)} - \mathbf{x}_k^{(\ell-1)}\|_2^2, \tag{26}$$

with $c_k := \frac{1}{\eta_k} - \frac{L_k}{2} > 0$.

**Step 2: Monotone decrease and square-summable steps.** Summing Equation (26) over $k = K, K-1, \ldots, 0$ telescopes the intermediate objectives:

$$\mathcal{J}_i(X^{(\ell)}) = \mathcal{J}_i\big(X^{(\ell,-1)}\big) \leq \mathcal{J}_i\big(X^{(\ell,K)}\big) - \sum_{k=0}^{K} c_k \|\mathbf{x}_k^{(\ell)} - \mathbf{x}_k^{(\ell-1)}\|_2^2.$$

Since $X^{(\ell,K)} = X^{(\ell-1)}$, we obtain the per-sweep descent inequality

$$\mathcal{J}_i(X^{(\ell)}) \leq \mathcal{J}_i(X^{(\ell-1)}) - \sum_{k=0}^{K} c_k \|\mathbf{x}_k^{(\ell)} - \mathbf{x}_k^{(\ell-1)}\|_2^2. \tag{27}$$

This implies monotone decrease, proving item (1). Moreover, by Assumption D.1(1), $\mathcal{J}_i$ is bounded below, so summing Equation (27) over $\ell = 1, 2, \ldots, T$ and letting $T \to \infty$ yields

$$\sum_{\ell \geq 1} \sum_{k=0}^{K} c_k \|\mathbf{x}_k^{(\ell)} - \mathbf{x}_k^{(\ell-1)}\|_2^2 \leq \mathcal{J}_i(X^{(0)}) - \inf \mathcal{J}_i < \infty.$$

Since each $c_k > 0$, it follows that $\sum_{\ell \geq 1} \sum_{k=0}^{K} \|\mathbf{x}_k^{(\ell)} - \mathbf{x}_k^{(\ell-1)}\|_2^2 < \infty$, and therefore $\|\mathbf{x}_k^{(\ell)} - \mathbf{x}_k^{(\ell-1)}\|_2 \to 0$ for every $k$, proving item (2).

**Step 3: Stationarity of accumulation points.** Let $\bar{X}$ be any accumulation point of $\{X^{(\ell)}\}_{\ell \geq 0}$. Then there exists a subsequence $\{\ell_j\}$ such that $X^{(\ell_j)} \to \bar{X}$ as $j \to \infty$. Fix any block index $k$. By the definition of $X^{(\ell,k)}$, the only blocks in which $X^{(\ell,k)}$ and $X^{(\ell)}$ can differ are $0, 1, \ldots, k$, hence

$$\|X^{(\ell,k)} - X^{(\ell)}\|_2 \leq \sum_{r=0}^{k} \|\mathbf{x}_r^{(\ell-1)} - \mathbf{x}_r^{(\ell)}\|_2.$$

By item (2), the right-hand side converges to 0 as $\ell \to \infty$, so along the subsequence we also have $X^{(\ell_j,k)} \to \bar{X}$.

Next, the block update Equation (15) can be rewritten as

$$\nabla_{\mathbf{x}_k} \mathcal{J}_i\big(X^{(\ell,k)}\big) = \frac{1}{\eta_k}\Big(\mathbf{x}_k^{(\ell-1)} - \mathbf{x}_k^{(\ell)}\Big).$$

By item (2), the right-hand side tends to 0 as $\ell \to \infty$, hence $\nabla_{\mathbf{x}_k} \mathcal{J}_i(X^{(\ell_j,k)}) \to 0$ as $j \to \infty$. Since $\mathcal{J}_i$ is continuously differentiable (Assumption D.1(2)) and $X^{(\ell_j,k)} \to \bar{X}$, we may pass to the limit to obtain $\nabla_{\mathbf{x}_k} \mathcal{J}_i(\bar{X}) = 0$. Because $k$ was arbitrary, this holds for all blocks $k = 0, \ldots, K$, and therefore $\nabla \mathcal{J}_i(\bar{X}) = 0$. This proves item (3).  □

## E. Experimental Settings

### E.1. Computed Tomography

The OrganCMNIST dataset (Yang et al., 2021; 2023) contains $23\,582$ abdominal CT images of size $64 \times 64$ pixels, split into $12\,975$ training, 2392 validation, and 8216 test images. We adopt the affine flow path $x_t = (1-t)x_0 + tx_1$ and parameterize the flow with a time-dependent UNet (Dhariwal & Nichol, 2021) consisting of approximately 8M parameters.

For D-Flow and MS-flow the initial estimate of $\mathbf{x}_0$ is computed following (Ben-Hamu et al., 2024), as

$$\mathbf{x}_0 = \sqrt{\beta}\,\mathbf{w}(0) + \sqrt{1-\beta}\,\mathbf{z},$$

where $\mathbf{z} \sim \mathcal{N}(0, I)$, and $\mathbf{w}(0) = \mathbf{w} + \int_1^0 v_\theta(\mathbf{w}(t), t)$ with $\mathbf{w}$ being the filtered back-projection computed from the observed data $\mathbf{y}$. For MS-Flow, by first propagating $\mathbf{x}_0$ with the given flow $v_\theta$, using the explicit Euler scheme, and then linearly interpolating between the $\mathbf{x}_0$ and the end-point of that trajectory. In doing so, the trajectory consistency loss is not equal to zero at the start. The initial $\mathbf{x}^*$ is computed as the endpoint of that trajectory.

The average run times in Table 2 were computed on the same machine with a single NVIDIA GeForce RTX 5090 GPU.

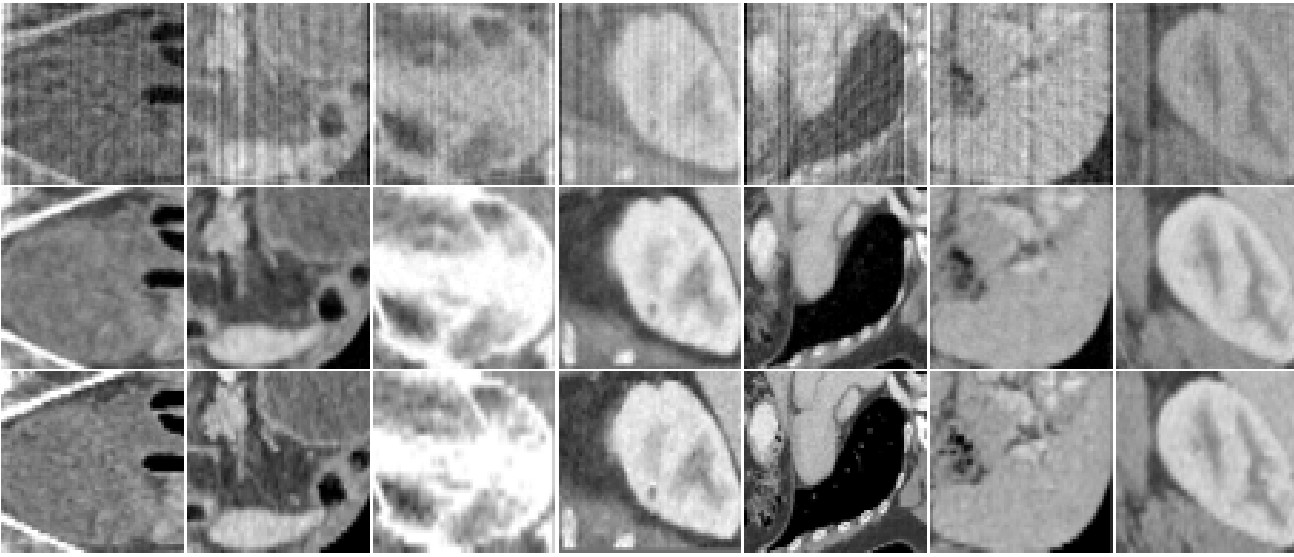

*Figure 6.* Example reconstructions on OrganCMnist. First row: filtered back-projection (baseline reconstruction). Second row: MS-Flow (our) reconstruction. Third row: Ground truth.

### E.2. Image recovery

CelebA dataset (Yang et al., 2015) consists of more than $200\,000$ images resized to $128 \times 128$ pixels. As the flow model, we use the pretrained model obtained from `https://github.com/annegnx/PnP-Flow`. The parameters for all methods are tuned on a small set of test images. The quality metrics are then evaluated on the first 50 images from the validation set.

*Table 5.* Hyperparameter settings across image restoration tasks. For Flow-Prior we give $(\eta, \lambda)$, for OT-ODE $(t_0, \gamma)$, for PnP-Flow $(\alpha)$, for D-Flow $(n_t, \lambda)$ and for MS-Flow $(n_t, L, \gamma, \alpha, \eta, \lambda)$

| Method | Image restoration tasks | | | |
| --- | --- | --- | --- | --- |
| | CT | Super-res | Gaussian Debl. | Box-inpaint |
| Flow-Priors | – | (0.01, 1e4) | (0.01, 1e4) | (0.01, 1e4) |
| OT-ODE | – | (0.2, constant) | (0.4, $\sqrt{t}$) | (0.1, $\sqrt{t}$) |
| PnP-Flow | – | (0.01) | (0.01) | (0.01) |
| D-Flow | (3, 0.05) | (3, 0.05) | (3, 1) | (3, 1) |
| MS-Flow | (6, 1, 0.01, 0.1, 5, 0.0001) | (6, 5, 0.01, 0.1, 5, 0.0001) | (6, 1, 0.01, 0.1, 5, 0.0001) | (12, 1, 0.01, 1, 5, 0.000) |

Similarly to above, for D-Flow and MS-flow the initial estimate of $\mathbf{x}_0$ is computed following (Ben-Hamu et al., 2024), as

$$\mathbf{x}_0 = \sqrt{\beta}\,\mathbf{w}(0) + \sqrt{1-\beta}\,\mathbf{z},$$

where $\mathbf{z} \sim \mathcal{N}(0, I)$, and $\mathbf{w}(0) = \mathbf{w} + \int_1^0 v_\theta(\mathbf{w}(t), t)$ with $\mathbf{w}$ computed from the observed data $\mathbf{y}$, depending on the given task (as the noisy observation $\mathbf{y}$ for Gaussian deblurring, by applying the adjoint of the forward operator to $\mathbf{y}$ for super-resolution, and as a random image for box-inpainting). The initial trajectory and the initial $\mathbf{x}^*$ are then computed in the same way as for Computed Tomography.

In Table 5 we provide the parameter choices for all methods and each restoration setting.

**Flow-Priors** (Zhang et al., 2024) We consider the hyperparameters $\eta$ and $\lambda$, corresponding to the step size for gradient descent and the weighting of the data-consistency term. We perform a grid search over $\eta \in \{0.001, 0.01, 0.1\}$ and $\lambda \in \{1 \times 10^2, 1 \times 10^3, 1 \times 10^4, 1 \times 10^5\}$. We use $N = 100$ discretization steps.

**OT-ODE** (Pokle et al., 2024) We consider the hyperparameters $t_0$ (initial time), learning rate type $\gamma$. We do a grid search over $t_0 \in \{0.1, 0.2, 0.3, 0.4\}$ and $\gamma \in \{\text{constant}, \sqrt{t}\}$. We again use $N = 100$ discretization steps.

**PnP-Flow** (Martin et al., 2025) We consider the hyperparameters $\alpha$ (exponent of learning rate) and search over $\alpha \in \{0.01, 0.1, 0.3, 0.5, 0.8, 1.0\}$. We use $N = 100$ discretization steps. Following Martin et al. (2025), we use the average of 5 evaluations of the flow field in every step.

### E.3. Further Results

Results in Table 6 investigate the stability over the optimization trajectory for D-Flow and MS-Flow. In particular, we compare the average highest PSNR over the optimization trajectory with the average PSNR of the converged trajectory (evaluated at the last iteration). As previously discussed, D-Flow often shows a comparably high maximum PSNR, but afterward it tends to overfit and deteriorate in performance. In comparison, MS-Flow shows a stable performance with small differences between the best PSNR and the PSNR of the converged image.

*Table 6.* Reconstruction quality comparison for CelebA: D-Flow vs. MS-Flow. We show the best PSNR over iterations and the PSNR of the converged image.

| Task | Method | PSNR ↑ | | SSIM ↑ | |
| --- | --- | --- | --- | --- | --- |
| | | Converged | Best | Converged | Best |
| Box Inpainting | D-Flow | 33.96 | 35.84 | 0.9792 | 0.9736 |
| | MS-Flow | 36.30 | 36.46 | 0.9799 | 0.9800 |
| Super-resolution | D-Flow | 34.67 | 34.69 | 0.9314 | 0.9322 |
| | MS-Flow | 36.46 | 36.46 | 0.9570 | 0.9570 |
| Gaussian Deblurring | D-Flow | 33.48 | 34.66 | 0.8696 | 0.9102 |
| | MS-Flow | 38.02 | 38.40 | 0.9587 | 0.9636 |

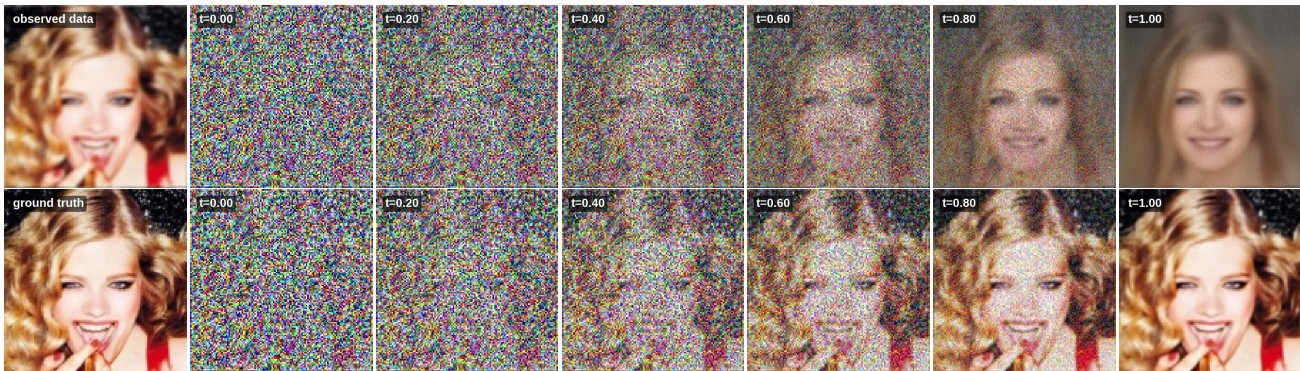

*Figure 7.* Top: initial shooting-point initialization. Bottom: converged shooting points after optimization. Results shown for Gaussian deblurring, alongside the measurements and ground-truth image.

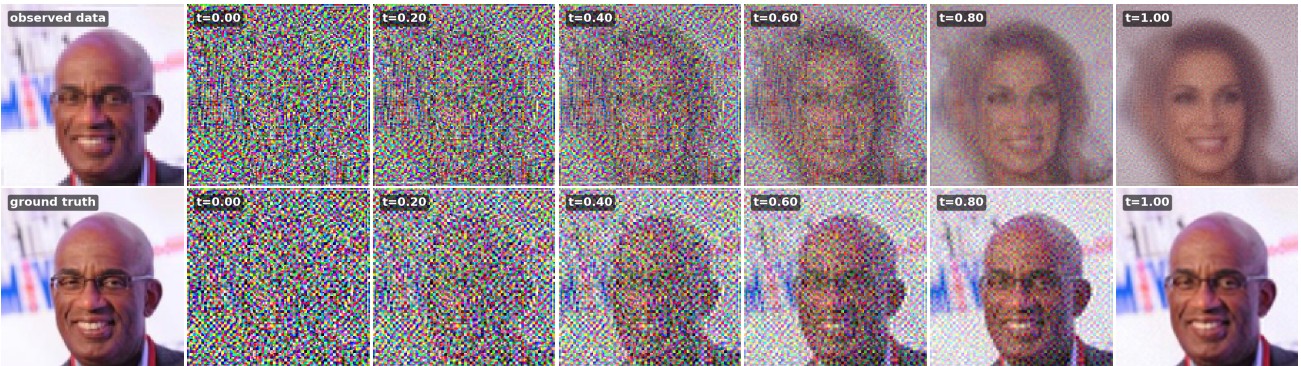

*Figure 8.* Top: initial shooting-point initialization. Bottom: converged shooting points after optimization. Results shown for super-resolution, alongside the measurements and ground-truth image.

## F. Application to Latent Flow Models

In latent flow models, the dynamics of the flow model are defined in latent space. We refer to the image space $\mathcal{X}$ and the latent space with $\mathcal{Z}$. We define the encoder at $E : \mathcal{X} \to \mathcal{Z}$ and the decoder as $D : \mathcal{Z} \to \mathcal{X}$. The trained flow model $v_\theta : \mathcal{Z} \times [0,1] \to \mathcal{Z}$ acts only in latent space. Sample generation is then performed via

$$\frac{d\mathbf{z}}{dt} = v_\theta(\mathbf{z}, t), \quad t \in [0,1], \mathbf{z}(0) = \mathbf{z}_0, \tag{28}$$

with our final sample as $\mathbf{x} = D(\mathbf{z}(1))$. So, the dynamics are fully in latent space, and only at the final step do we decode back to the image space. For Latent MS-Flow, we introduce latent control points

$$\{\mathbf{z}_0, \dots, \mathbf{z}_K\}, 0 = t_0 < \cdots < t_K = 1 \tag{29}$$

and a terminal variable $\mathbf{z}_*$ in latent space. The augmented objective is then

$$Z^{i+1} = \arg \min_{[\mathbf{z}_0, \dots, \mathbf{z}_K]} \frac{\alpha}{2} \|\mathbf{z}_*^i - \mathbf{z}_K\|^2 + \lambda \mathcal{R}(\mathbf{z}_0) + \frac{\gamma}{2} \sum_{k=1}^{K} \|\mathbf{z}_k - F_{k-1,k}(\mathbf{z}_{k-1})\|^2 \tag{30a}$$

$$\mathbf{z}_*^{i+1} = \arg \min_{\mathbf{z}_*} \Phi(D(\mathbf{z}_*)) + \frac{\alpha}{2} \|\mathbf{z}_* - \mathbf{z}_K^{i+1}\|^2. \tag{30b}$$

As the terminal loss $\Phi$ is usually defined in image space, the data-consistency step requires backpropagation of the decoder.

### F.1. Implementation Details

Stable Diffusion 3.5[1] is trained via classifier-free guidance (CFG) (Ho & Salimans, 2022). We use a CFG weight of 2.0 and use the prompt "a high quality image of a face." as the conditional text input, both for FlowDPS (Kim et al., 2025) and MS-Flow. For MS-Flow, we initialize $\mathbf{z}_*^0$ as a zero vector. We then initialize the shooting points $[\mathbf{z}_0, \dots, \mathbf{z}_K]$ as a linear interpolation between $\mathbf{z}_0 \sim \mathcal{N}(0, I)$ and $\mathbf{z}_*^0$. We use $K = 29$ shooting points, which is the number of integration steps usually chosen for Stable Diffusion 3.5. We use 10 inner steps and 25 alternating minimization steps in total, with $\alpha = \gamma = 0.06$.

## G. Connection to MAP Estimation

We motivate the MS-Flow objective from an optimization (multiple-shooting) viewpoint. However, one can obtain the same objective also as MAP estimation over a joint posterior. For this, we consider the trajectory $\mathbf{x}_{0:K} = (\mathbf{x}_0, \dots, \mathbf{x}_K)$ with flow dynamics $\mathbf{x}_k = F_{k-1,k}(\mathbf{x}_{k-1})$. We relax the dynamics with local Gaussian penalties, i.e.,

$$p(\mathbf{x}_k|\mathbf{x}_{k-1}) = \mathcal{N}(\mathbf{x}_k; F_{k-1,k}(\mathbf{x}_{k-1}), \gamma^{-1}\mathbf{I}).$$

Further, we assume $p(\mathbf{x}_*|\mathbf{x}_K) = \mathcal{N}(\mathbf{x}_*|\mathbf{x}_K, \alpha^{-1}\mathbf{I})$ and $p(\mathbf{x}_*|\mathbf{x}_{0:K}) = p(\mathbf{x}_*|\mathbf{x}_K)$. Then, we can consider the joint density

$$p(\mathbf{x}_*, \mathbf{x}_{0:K}|y) \propto \exp(-\Phi(\mathbf{x}_*; y)) \exp(-\frac{\alpha}{2}\|\mathbf{x}_* - \mathbf{x}_K\|^2) \exp(-\lambda R(\mathbf{x}_0)) \prod_{k=1}^{K} \exp\left(-\frac{\gamma}{2}\|\mathbf{x}_k - F_{k-1,k}(\mathbf{x}_{k-1})\|^2\right), \quad (31)$$

where taking the negative-log likelihood gives exactly the MS-Flow objective (7):

$$\underset{\mathbf{x}_*, \mathbf{x}_{0:K}}{\arg\min} - \log p(\mathbf{x}_*, \mathbf{x}_{0:K}|y) = \underset{\mathbf{x}_*, \mathbf{x}_{0:K}}{\arg\min} \left\{ \Phi(\mathbf{x}_*; y) + \frac{\alpha}{2}\|\mathbf{x}_* - \mathbf{x}_K\|^2 + \lambda R(\mathbf{x}_0) + \frac{\gamma}{2}\sum_{k=1}^{K}\|\mathbf{x}_k - F_{k-1,k}(\mathbf{x}_{k-1})\|^2 \right\}. \quad (32)$$

**Extension to Diffusion Models**   In particular, this viewpoint naturally extends to diffusion models (Ho et al., 2020). In contrast to the deterministic flow dynamics, diffusion models define a stochastic forward process that progressively maps data to a simple latent distribution. Note, that the usual notation of diffusion models reverses the time directions, i.e., $\tilde{\mathbf{x}}_0$ denotes clean data and $\tilde{\mathbf{x}}_K$ the latent variables. We will use $\tilde{\mathbf{x}}$ to denote the states in the diffusion model. The forward process is given by the Markov chain

$$q(\tilde{\mathbf{x}}_k \mid \tilde{\mathbf{x}}_{k-1}) = \mathcal{N}\left(\tilde{\mathbf{x}}_k; \sqrt{1 - \beta_k}\,\tilde{\mathbf{x}}_{k-1}, \beta_k\mathbf{I}\right), \quad (33)$$

for a noise schedule $\{\beta_k\}_{k=1}^{K}$. The corresponding learned reverse process is parameterized as

$$p_\theta(\tilde{\mathbf{x}}_{k-1} \mid \tilde{\mathbf{x}}_k) = \mathcal{N}\left(\tilde{\mathbf{x}}_{k-1}; \mu_\theta(\tilde{\mathbf{x}}_k, k), \sigma_k^2\mathbf{I}\right), \quad (34)$$

where $\mu_\theta$ is parametrized by the trained diffusion model, which induces a generative trajectory starting from $\tilde{\mathbf{x}}_K \sim \mathcal{N}(\mathbf{0}, \mathbf{I})$. We now consider a lifted posterior over the entire reverse-time diffusion trajectory $\tilde{\mathbf{x}}_{0:K}$:

$$p_\theta(\mathbf{x}_*, \tilde{\mathbf{x}}_{0:K} \mid y) \propto \exp(-\Phi(\mathbf{x}_*; y)) \exp(-\frac{\alpha}{2}\|\mathbf{x}_* - \tilde{\mathbf{x}}_K\|^2) \exp(-\lambda R(\tilde{\mathbf{x}}_K)) \prod_{k=1}^{K} p_\theta(\tilde{\mathbf{x}}_{k-1} \mid \tilde{\mathbf{x}}_k), \quad (35)$$

under similar assumption as above, i.e., $p(\mathbf{x}_*|\tilde{\mathbf{x}}_0) = \mathcal{N}(\mathbf{x}_*|\tilde{\mathbf{x}}_0, \alpha^{-1}\mathbf{I})$ and $p(\mathbf{x}_*|\tilde{\mathbf{x}}_{0:K}) = p(\mathbf{x}_*|\tilde{\mathbf{x}}_0)$. Taking the negative log yields the corresponding MAP objective over the full diffusion trajectory:

$$\underset{\mathbf{x}_*, \tilde{\mathbf{x}}_{0:K}}{\arg\min} - \log p(\mathbf{x}_*, \tilde{\mathbf{x}}_{0:K} \mid y) = \underset{\mathbf{x}_*, \tilde{\mathbf{x}}_{0:K}}{\arg\min} \left\{ \Phi(\mathbf{x}_*; y) + \frac{\alpha}{2}\|\mathbf{x}_* - \tilde{\mathbf{x}}_0\|^2 + \lambda R(\tilde{\mathbf{x}}_K) + \sum_{k=1}^{K}\frac{1}{2\sigma_k^2}\|\tilde{\mathbf{x}}_{k-1} - \mu_\theta(\tilde{\mathbf{x}}_k, k)\|^2 \right\}. \quad (36)$$

---

[1]https://huggingface.co/stabilityai/stable-diffusion-3.5-medium

## H. Salt-and-Pepper Noise

We consider salt-and-pepper noise, which allows us to highlight the applicability of MS-Flow to non-smooth convex objective functions. The noise model is given by $y_i^\delta = y_i$ with probability $1 - p$, $y_i^\delta = 0$ with probability $p/2$, and $y_i^\delta = 1$ with probability $p/2$. The corresponding data-fidelity term is given by the $L_1$-Norm instead of the mean-squared-error, which is handled by D-Flow with backpropagation (the same as for the case of the $L_2$-norm), and with linearized ADMM (discussed in Appedix I) for MS-Flow (which is equal to the proximal operator of the data fidelity term).

For the forward operator we consider the Gaussian blur with kernel size 15 and intensity $\sigma = 1.5$. We consider two noise regimes: low noise (with $p = 0.05$), and a high noise regime (with $p = 0.1$). The results in Table 7 show that MS-Flow substantially outperforms DFlow in this setting, demonstrating its robustness to both high noise and non-differentiable regularization.

*Table 7.* Reconstruction quality for salt-and-pepper noise, with Gaussian deblurring on CelebA comparing D-Flow and MS-Flow.

| Task | Method | PSNR (↑) | SSIM (↑) | LPIPS (↓) |
|------|--------|----------|----------|-----------|
| Low noise ($p = 0.05$) | D-Flow | 33.26 | 0.927 | 0.042 |
| | MS-Flow | 37.33 | 0.957 | 0.013 |
| High noise ($p = 0.1$) | D-Flow | 28.84 | 0.846 | 0.074 |
| | MS-Flow | 30.23 | 0.857 | 0.094 |

## I. ADMM Formulation of MS-Flow

We present an alternative method for solving the MS-Flow objective. Instead of using the splitting in (7), we introduce an ADMM splitting. For the ADMM splitting, we first discretize the ODE in (6), which results in the following constrained optimization problem

$$\arg\min_{\mathbf{x_0}} \Phi(\mathbf{x}_K) + \mathcal{R}(\mathbf{x}_0) \tag{37}$$

$$\text{s.t. } \mathbf{x}_{k+1} = \mathbf{x}_k + \Delta t v_\theta(\mathbf{x}_k, t_k), \ k = 0, \dots, K-1, \tag{38}$$

here, we use a simple Euler discretization for simplicity, but other choices are possible as well. We then introduce an additional variable $x^*$ and consider

$$\arg\min_{\mathbf{x}_*, \mathbf{x}_{0:K}} \Phi(\mathbf{x}_*) + \mathcal{R}(\mathbf{x}_0), \tag{39}$$

$$\text{s.t. } \mathbf{x}_* - \mathbf{x}_K = 0 \tag{40}$$

$$c_k(\mathbf{x}_{0:K}) := \mathbf{x}_{k+1} - \mathbf{x}_k - \Delta t v_\theta(\mathbf{x}_k, t_k) = 0, \tag{41}$$

$$k = 0, \dots, K-1. \tag{42}$$

We define the augmented Lagrangian

$$\mathcal{L}_{\gamma,\alpha}(\mathbf{x}_*, \mathbf{x}_{0:K}, \lambda_{0:K-1}, p) = \Phi(\mathbf{x}_*) + \mathcal{R}(\mathbf{x}_0) + \sum_{k=0}^{K-1} \lambda_k^T c_k(\mathbf{x}_{0:K}) + \frac{\gamma}{2} \sum_{k=0}^{K-1} \|c_k(\mathbf{x}_{0:K})\|^2 + p^T(\mathbf{x}_* - \mathbf{x}_K) + \frac{\alpha}{2}\|\mathbf{x}_* - \mathbf{x}_K\|^2.$$

We use alternating minimization, where we first do an update with respect to $\mathbf{x}_*$, then an update for the trajectory $\mathbf{x}_{0:K}$, and finally an update for the dual parameters $\lambda_{0:K-1}$ and $p$. We have $\gamma$ and $\alpha$ as penalty parameters, which can be set automatically, e.g., according to the rule in Section 3.4.1 in (Boyd et al., 2011). The updates are as follows.

**Update $\mathbf{x}_*$** We have to solve

$$\mathbf{x}_*^{(n+1)} = \arg\min_{\mathbf{x}_*} \Phi(\mathbf{x}_*) + p^{(n)^T}(\mathbf{x}_* - \mathbf{x}_K^{(n)}) + \frac{\alpha}{2}\|\mathbf{x}_* - \mathbf{x}_K^{(n)}\|^2 \tag{43}$$

$$= \arg\min_{\mathbf{x}_*} \Phi(\mathbf{x}_*) + \frac{\alpha}{2}\|\mathbf{x}_* - (\mathbf{x}_K^{(n)} - \frac{1}{\alpha}p^{(n)})\|^2 \tag{44}$$

$$= \text{prox}_{\Phi/\alpha}(\mathbf{x}_K^{(n)} - \frac{1}{\alpha}p^{(n)}) \tag{45}$$

**Update $\mathbf{x}_{0:K}$**  The trajectory constraint is in this update. Here, we have to solve

$$\min_{\mathbf{x}_{0:K}} \mathcal{R}(\mathbf{x}_0) + \sum_{k=0}^{K-1} \lambda_k^{(n)^T} c_k(\mathbf{x}_{0:K}) + \frac{\gamma}{2} \sum_{k=0}^{K-1} \|c_k(\mathbf{x}_{0:K})\|^2 - p^{(n)^T} \mathbf{x}_K + \frac{\alpha}{2} \|\mathbf{x}_*^{(n+1)} - \mathbf{x}_K\|^2 \tag{46}$$

which can be written as

$$\min_{\mathbf{x}_{0:K}} \mathcal{R}(\mathbf{x}_0) + \sum_{k=0}^{K-1} \lambda_k^{(n)^T} c_k(\mathbf{x}_{0:K}) + \frac{\gamma}{2} \sum_{k=0}^{K-1} \|c_k(\mathbf{x}_{0:K})\|^2 + \frac{\alpha}{2} \|\mathbf{x}_K - (\mathbf{x}_*^{(n+1)} + \frac{1}{\alpha} p^{(n)})\|^2 \tag{47}$$

For $k = K$ we get:

$$\min_{\mathbf{x}_K} \lambda_{K-1}^{(n)^T} c_{K-1}(\mathbf{x}_{0:K}) + \frac{\gamma}{2} \|c_{K-1}(\mathbf{x}_{0:K})\|^2 + \frac{\alpha}{2} \|\mathbf{x}_K - (\mathbf{x}_*^{(n+1)} + \frac{1}{\alpha} p^{(n)})\|^2 \tag{48}$$

For $0 < k < K$ we get:

$$\min_{\mathbf{x}_k} \lambda_{k-1}^{(n)^T} c_{k-1}(\mathbf{x}_{0:K}) + \frac{\gamma}{2} \|c_{k-1}(\mathbf{x}_{0:K})\|^2 + \frac{\gamma}{2} \|c_k(\mathbf{x}_{0:K})\|^2 + \lambda_k^{(n)^T} c_k(\mathbf{x}_{0:K}) \tag{49}$$

and for $k = 0$ we get

$$\min_{\mathbf{x}_0} \mathcal{R}(\mathbf{x}_0) + \lambda_0^{(n)^T} c_0(\mathbf{x}_{0:K}) + \frac{\gamma}{2} \|c_0(\mathbf{x}_{0:K})\|^2 \tag{50}$$

**Update dual variables**  These are again simple

$$\lambda_k^{(n+1)} = \lambda_k^{(n)} + \gamma c_k(\mathbf{x}_{0:K}^{(n+1)}), \ k = 0, \ldots, K-1 \tag{51}$$

$$p^{(n+1)} = p^{(n)} + \alpha(\mathbf{x}_*^{(n+1)} - \mathbf{x}_K^{(n+1)}) \tag{52}$$

## J. Ablation: Number of inner iterations vs PSNR

Here, we examine the reconstruction performance of MS-Flow as a function of the number of inner iterations (parameter $L$ in Algorithm 1). Increasing the number of iterations can improve performance, for example, in box inpainting, but it also increases computational time approximately linearly. Moreover, these performance gains do not necessarily transfer across different inverse problems. We present results in Table 8.

*Table 8.* Reconstruction quality for box inpainting on CelebA comparing D-Flow and MS-Flow, with respect to the number of inner iterations

| Method | PSNR ($\uparrow$) | SSIM ($\uparrow$) | LPIPS ($\downarrow$) |
|---|---|---|---|
| D-Flow | 34.37 | 0.968 | 0.015 |
| MS-Flow ($L = 1$ iteration) | 34.68 | 0.974 | 0.015 |
| MS-Flow ($L = 5$ iterations) | 35.62 | 0.976 | 0.013 |

