# OpenReview forum: "Trajectory Stitching for Solving Inverse Problems with Flow-Based Models"
_ICML.cc/2026/Conference — ICML 2026 regular_

### Official Review · Reviewer_9cBy · 2026-02-23

**Soundness:** 3
**Presentation:** 3
**Significance:** 2
**Originality:** 2
**Overall Recommendation:** 3
**Confidence:** 4

**Summary:**

The authors propose a new method for solving inverse problems with flow models. Prior work has explored solving inverse problems via latent-space optimisation, where the initial noise is directly optimised through the full generative trajectory. However, differentiating through the entire trajectory incurs significant memory costs and can lead to numerical instability. To address this problem, the authors split the flow trajectory into multiple segments and optimise intermediate states via coordinate descent. This avoids full backpropagation and results in substantially reduced memory usage.  The method is evaluated on several inverse problems,  including computed tomography and super-resolution demonstrating competitive reconstruction performance.

**Compliance With Llm Reviewing Policy:**

Affirmed.

**Final Justification:**

The authors addressed several of my concerns during the rebuttal. However, the method still appears to underperform on more challenging tasks (4 super resolution non-linear deblurring). I therefore decide to maintain my score.

**Key Questions For Authors:**

- Could the authors further clarify the argument in Section 4.1, particularly how the Grönwall-based sensitivity bound is affected by the segmented trajectory formulation? Additionally, could they clarify the novelty and role of Proposition 4.1?
- Could the authors provide runtime comparisons for all methods in the image recovery tasks ? what resolution are the images used in those tasks ?
- Could the authors include perception based metrics such as LPIPS and FID in their evaluation ?
- Could the evaluation be extended to a larger number of images (e.g at least 100 per setting) and real-world datasets such as ImageNet ?
- Is the method applicable to more challenging inverse problem settings, such as 128 by 128 box inpainting or 4 super resolution ?

**Limitations:**

yes

**Strengths And Weaknesses:**

**Strengths**
- The paper is well written and easy to follow. The idea is clearly motivated by prior work.
- The method demonstrates consistent improvements over D-Flow, the main competing approach, in reconstruction quality while     substantially reducing memory usage.
- The experimental evaluation includes computed tomography experiments, extending beyond standard image restoration tasks.

**Weaknesses**
- Despite the method being more efficient than D-Flow, it still requires running iterative optimisation over multiple variables, which can become computationally prohibitive for high-resolution or large-scale models.
- Section 4.1 provides intuition based on Grönwall’s inequality however, it is not entirely clear how this stability bound translates to the stability of the full segmented trajectory, since local errors may still accumulate across segments. Providing a more rigorous argument would strengthen the theoretical justification.
- In Section 4.2, the authors prove that under their set of assumptions the Gauss-Seidel iterations of the corresponding subproblem converge. However, this result appears to be a classical coordinate descent convergence guarantee and is largely independent of the inverse problem formulation.
- The experimental protocol is somewhat limited. Specifically, the degradations (32 by 32 box inpainting, 2 super-resolution) are significantly milder than those used in recent benchmarks such as DAPS [1] and DPS [2]. Moreover, the evaluation is conducted on 50 images only. Furthermore, the lack of large-scale datasets such as  ImageNet makes it difficult to assess the scalability of the method.
- The experimental results for image recovery tasks report only PSNR and SSIM, which do not fully capture perceptual quality.
- The authors include experiments with latent flow models however, the evaluation uses only 10 FFHQ images, a single inverse problem and one competing method. This setting is insufficient to assess the effectiveness of the approach.


**References**
- [1] Zhang, B., Chu, W., Berner, J., Meng, C., Anandkumar, A.,
and Song, Y. Improving diffusion inverse problem solving
with decoupled noise annealing. In Proceedings of the
Computer Vision and Pattern Recognition Conference, pp.
20895–20905, 2025.
- [2] Chung, H., Kim, J., Mccann, M. T., Klasky, M. L., and
Ye, J. C. Diffusion posterior sampling for general noisy
inverse problems. In The Eleventh International Conference on Learning Representations, 2023.

---

> ### Author Rebuttal · Authors · 2026-03-31
>
> We thank you for your careful review feedback. We appreciate that you found the paper well written and easy to follow, that you viewed the idea as well motivated by prior work, and that you recognized the substantial memory reduction. We clarified the interpretation of the stability and convergence results, expanded the evaluation to larger test sets ($100$ images for all settings), added LPIPS, added harder tasks including $4\times$ super-resolution and non-linear deblurring, and added a compute-fair comparison based on PSNR vs inference time in the revised paper. We hope that you find our responses satisfactory, and that you will consider revising your score.
>
> **Regarding the Grönwall-based stability discussion**
>
> Thank you for this important question. We clarify that the role of Section 4.1 is to provide **conditioning intuition**, rather than a global error-propagation theorem.
>
> The key point is the following. In single shooting, the terminal state depends on the composition of the flow over the entire interval $[0,1]$, so perturbations in the optimized variable can be amplified by the full-horizon sensitivity. Under an $L$-Lipschitz vector field, this gives the standard factor $e^{Lt}$, and at terminal time $e^L$. In MS-Flow, we instead introduce intermediate variables $x_0,\dots,x_K$ and penalize **local defects**
>
> $$
> \|x_{k+1} - F_{k,k+1}(x_k)\|^2,
> $$
>
> where each segment has length $\Delta_k$. Local sensitivity scales with $e^{L\Delta_k}$.
>
> While local errors could accumulate if propagated forward, MS-Flow does not optimize a single trajectory. Each segment has its own state and is corrected locally through the defect penalties and the block updates. In other words, errors are not forced to propagate unchecked through a length-1 composition; they are repeatedly re-optimized and re-stitched at every segment. We added this clarification to the revised paper to make the role of the Grönwall argument more precise.
>
> **Regarding Proposition 4.1 and the Gauss-Seidel analysis**
>
> Thank you for this insightful point. Proposition 4.1 is not intended as a new generic theorem for coordinate descent. We added wording to make this explicit in the revised paper. Its role is to show that, for the **MS-Flow trajectory subproblem**, cyclic block updates:
> 1. decrease the objective monotonically,
> 2. and converge to stationary points.
>
> MS-Flow is an **inference-time optimization method**, and stability of the trajectory updates is critical. In addition, Remark 4.2 explains how this extends to the **inexact / Jacobian-free updates** used in practice via sufficient decrease. We added this clarification to the revised paper so that the proposition is positioned as a method-specific guarantee for the optimization subroutine, rather than as a general novelty claim.
>
> **Regarding the experimental protocol and perceptual quality**
>
> Thank you for these suggestions. Following your comments, we expanded the evaluation in several ways in the revised paper:
> - we evaluate on **100 images** for all experiments,
> - we added **LPIPS** as a perceptual metric, and
> - we added a compute-fair **PSNR vs. inference-time** comparison.
>
> For the CelebA restoration tasks, we now report:
>
> | Method | Gaussian Deblurring PSNR | Gaussian Deblurring SSIM | Gaussian Deblurring LPIPS | Box-Inpainting PSNR | Box-Inpainting SSIM | Box-Inpainting LPIPS | SR PSNR | SR SSIM | SR LPIPS |
> |---|---:|---:|---:|---:|---:|---:|---:|---:|---:|
> | Flow-Prior | 36.12 | 0.961 | 0.009 | 36.79 | 0.984 | 0.011 | 34.47 | 0.946 | 0.015 |
> | OT-ODE | 37.51 | 0.969 | 0.015 | 34.26 | 0.975 | 0.013 | 35.21 | 0.956 | 0.008 |
> | PnP-Flow | 36.19 | 0.962 | 0.049 | 36.43 | 0.985 | 0.011 | 34.73 | 0.948 | 0.059 |
> | D-Flow | 33.17 | 0.890 | 0.029 | 33.96 | 0.979 | 0.012 | 34.36 | 0.942 | 0.019 |
> | **MS-Flow (Ours)** | **38.02** | 0.958 | **0.009** | 36.30 | 0.980 | 0.015 | **36.46** | **0.957** | 0.023 |
>
>
> **Regarding harder tasks**
>
> Thank you for this important suggestion. Following your comment, we added both **$4\times$ super-resolution** and **non-linear deblurring**.
>
> | Method | $4\times$ SR PSNR | $4\times$ SR SSIM | $4\times$ SR LPIPS | Non-linear Deblurring PSNR | Non-linear Deblurring SSIM | Non-linear Deblurring LPIPS |
> |---|---:|---:|---:|---:|---:|---:|
> | Flow-Prior | 27.27 | 0.796 | 0.094 | 21.84 | 0.490  |  0.318  |
> | OT-ODE | 28.18 | 0.834 | 0.052 | 21.95 | 0.626 | 0.118 |
> | PnP-Flow | 28.38 | 0.840 | 0.168 | 22.19  | 0.643  | 0.263 |
> | D-Flow | 29.93 | 0.873 | 0.062 | 26.41 |  0.682 | 0.069 |
> | **MS-Flow (Ours)** | 28.52 | 0.838 | 0.091 | 21.93 | 0.606 | 0.284 |
>
> We also clarified that the latent SD 3.5 experiment is already a non-linear inverse problem setting, because the data-consistency step is enforced through the nonlinear decoder. On nonlinear deblurring, MS-Flow achieves performance comparable to PnP-Flow and OT-ODE but does not match D-Flow. It seems that on this task being constrained to directly follow the trajectory is an advantage for D-Flow.

---

> > ### Author Rebuttal · Reviewer_9cBy · 2026-04-02
> >
> > I thank the authors for their response, in particular for the clarifications regarding the theoretical aspects of the paper and for extending the experiments to a larger set of images. However, I choose to maintain my score, as some of my concerns have not been fully addressed.
> >
> > - (W, Q2) The efficiency of the method relative to other baselines remains unclear.
> > - (Q3, Q4) While the authors presented results on a larger set of images and more challenging degradations, the method appears to underperform in these settings. In particular, performance degrades on the standard benchmark degradation (4× super-resolution). Additionally, FID scores were not reported.
> >
> > - W6) My concern regarding the limited evaluation on the latent flow model was not resolved.
> >
> > For these reasons, I choose to maintain my score.

---

> > > ### Author Response · Authors · 2026-04-02
> > >
> > > We thank you for your engagement and confirming that some of your concerns have been resolved. We are also thankful for the added comments which we now address point-by-point. We hope that you find our responses satisfactory, and that you will consider revising your score.
> > >
> > > ---
> > >
> > > **Regarding (W, Q2)**
> > >
> > >
> > > We thank you for raising the efficiency question. We agree that raw wall-clock across all baselines is not direct by design: Flow-Prior, OT-ODE, and PnP-Flow are forward-only guided sampling methods with fixed budgets, whereas D-Flow and MS-Flow are optimization-based inference methods. \
> > > Therefore, the relevant and direct efficiency comparison is MS-Flow vs. D-Flow. Complexity wise, D-Flow scales as $O(U n_t (C_{fwd} + C_{vjp}))$ because it differentiates through the full trajectory, while Jacobian-free MS-Flow scales as $O(T L K C_{fwd})$ and avoids backward-through-flow in the trajectory updates. Here, $U$ denotes the number of outer optimization steps in D-Flow, $n_t$ the number of discretization time points used per step, $C_{fwd}$ the cost of one forward flow evaluation, $C_{vjp}$ the cost of one backward-through-flow vector-Jacobian product, $T$ the number of shooting iterations in MS-Flow, $L$ the number of gradient updates per shooting iteration, and $K$ the number of shooting intervals or states updated per iteration. In the forward-only baselines, $S$ denotes the number of time steps, $C_{dc}$ the cost of one data-consistency update, and $r$ the number of repeated forward evaluations per step in PnP-Flow ($r=5$ in their setting).
> > >
> > > In our measurement we follow the paper settings with $T=1000, L=1, n_t=3$ for MS-Flow and 20 outer $\times$ 20 inner iterations with step-size search for D-Flow. We see that MS-Flow required 3000 forward evaluations and no backward-through-flow passes and ran in 14.1s, whereas D-Flow required about 5000 forward plus 5000 backward-through-flow evaluations and ran in 26.3s, i.e. MS-Flow is about 1.8x faster. Thus, while the forward-only baselines are expected to be faster in raw wall-clock by design, MS-Flow is substantially more efficient than the relevant optimization-based baseline D-Flow and offers a compelling, improved efficiency. We added this important discussion to the revised paper.
> > >
> > > | Method | Inference type | Theoretical complexity | Practical budget used in comparison | Executable runtime |
> > > |---|---|---|---|---:|
> > > | Flow-Prior | forward-only guidance | $O(S (C_{fwd} + C_{dc}))$ | 100 forward evaluations | 24.3s |
> > > | OT-ODE | forward-only guidance | $O(S (C_{fwd} + C_{dc}))$ | 100 forward evaluations | 5.19s |
> > > | PnP-Flow | forward-only guidance | $O(S (r C_{fwd} + C_{dc}))$ | 500 forward evaluations | 5.97s |
> > > | D-Flow | optimization, single shooting | $O(U n_t (C_{fwd} + C_{vjp}))$ | 5000 forward + 5000 backward-through-flow evaluations | 26.3s |
> > > | MS-Flow (Ours) | optimization, multiple shooting | $O(T L K C_{fwd})$ | 3000 forward evaluations, no backward-through-flow | 14.1s |
> > >
> > > **Regarding (Q3, Q4)**
> > >
> > > Thank you for your comment. To accommodate it, and following the practice in image reconstruction literature (see e.g. [1]), we focus on image level metrics. We therefore evaluate distortion with PSNR and SSIM, and perceptual quality with LPIPS.
> > >
> > > Additionally, we now show results (following the response to Reviewer uhNR) on non-linear phase retrieval, both in-distribution (CelebA) and out-of-distribution (AFHQ-cat). The results, reported in the Table below, show that MS-Flow is more robust to out-of-distribution evaluation, further highlighting the merit of our MS-Flow. We added these results to the paper. Thank you.
> > >
> > >
> > >
> > > | Phase Retrieval | PSNR ($\uparrow$) | SSIM ($\uparrow$) | LPIPS ($\downarrow$) | Runtime [sec per img] ($\downarrow$) |
> > > | -------- | -------- | -------- |-------- |-------- |
> > > | *CelebA:* D-Flow      | 25.12 | 0.656 | 0.193  | 93   |
> > > | *CelebA:* MS-Flow (Ours) | 24.08 | 0.691 | 0.187  | 58   |
> > > | *AFHQ-cat:* D-Flow   | 19.33  |0.409   |0.502   | 93 |
> > > | *AFHQ-cat:* MS-Flow (Ours) | 23.70  | 0.689  | 0.269  | 58 |
> > >
> > >
> > > **Regarding (W6):**  Due to space limitation we responded to this point to our response to *Reviewer 6AMv*. We are sorry that we did not reference to it in our rebuttal.
> > > Specifically, in our rebuttal, we included $100$ images for the test set and evalute against FLAIR as an additional latent space method, further highlighting the effectiveness of our MS-Flow, and we added these results to the revised paper. Thank you.
> > >
> > >
> > > Reference:
> > >
> > > [1] Martin, Ségolène, et al. "PNP-FLOW: Plug-And-Play Image Restoration with Flow Matching." International Conference on Learning Representations (ICLR), 2025.
> > >
> > > ---
> > >
> > > We would like to conclude by thanking you for your engagement and comments, and we hope that our responses to all your concerns are satisfactory.
> > >
> > > With best regards, \
> > > The Authors.

---

### Official Review · Reviewer_uhNR · 2026-03-06

**Soundness:** 2
**Presentation:** 3
**Significance:** 2
**Originality:** 3
**Overall Recommendation:** 4
**Confidence:** 4

**Summary:**

The paper presents **MS-Flow**, a multiple-shooting framework for solving inverse problems using flow-based generative priors. The authors replace the single-shooting optimization (directly optimizing initial noise) like D-Flow-based, to an approach that optimizes a sequence of intermediate latent states. By decomposing the generative trajectory into shorter segments and employing block-coordinate descent with trajectory-matching penalties, the method aims to decouple the flow dynamics (prior) from the data-consistency term. The primary claimed advantages are constant memory consumption relative to trajectory length and improved numerical stability over recent flow-based solvers like D-Flow.

**Compliance With Llm Reviewing Policy:**

Affirmed.

**Final Justification:**

The paper presents a novel approach that leverages **multiple-shooting techniques** to optimize trajectories in flow-matching solvers, thereby improving the **memory footprint** for solving inverse problems with these flow models. This technical contribution, combined with a solid theoretical analysis, represents a clear strength. Also, the authors did a good job in the rebuttal, improving the latent diffusion experiment. I want to thank the authors for running the new experiments.

However, the experimental validation is currently insufficient to explain the method's advantages:
* **Performance Gap:** While effective in simple scenarios, the methodology consistently underperforms compared to baselines in more complex settings.
* **Lack of Justification of why MS-Flow:** Although achieving State-of-the-Art (SOTA) results is not strictly necessary for acceptance, the authors could not provide enough justification for why **MS-flow** should be preferred over existing models if it does not offer a performance or efficiency trade-off in these harder cases. They did a good job in the rebuttal, though, and the OOD experiment partially does this, but from my point of view, it is not enough.

Consequently, while the idea is interesting and I support the acceptance over the rejection, the empirical section's weaknesses keep this submission at a 4, as the advantages of MS-flow in challenging inverse problems are not clear.

**Key Questions For Authors:**

The key questions are

1) A convincing discussion of how MS-Flow differs from [2] and why it would yield better solutions would strengthen the novelty claim (see Weakness 1 a).
2) Weakness 2
3) Weakness 3, in particular the sample size.
4) Weakness 5, the analysis of the PSNR as a function of inference-time (not only memory consumption). Given that MS-Flow has worse time complexity than D-Flow for $L>1$, a PSNR vs. inference time plot is essential to fairly evaluate whether the performance gains justify the additional compute

In particular, point 3 is a very important point, given that the sample size of 10 for the latent model is not representative. Providing results over at least 100 images for the main experiments, and at least 50 for the latent model experiment, would be necessary to increase the score.

**Limitations:**

Yes.

**Strengths And Weaknesses:**

## Strengths
* **Novel Formulation:** The application of multiple-shooting techniques to the optimizing the trajectory of flow models is an interesting and well-motivated approach to solving the memory and conditioning bottlenecks of methods that backpropagate through the trajectory.
* **Memory Efficiency:** The method successfully achieves constant memory scaling with respect to the number of timesteps, which is a significant practical advantage, particularly for high-resolution imaging.
* **Theoretical analysis:** The paper provides a solid theoretical analysis of the convergence of the proposed alternating minimization scheme and the stability benefits of local dynamics enforcement.
* **Experiments:** The experiments showcase the advantages of MS-flow over it competitors.
* **Writing:** The paper is well-written, with a coherent flow.

## Weaknesses

1. **Missing Related Reference**
There are several references that are missing, and are somehow related.

- a) **Methods that related directly:** Please discuss and cite [1],[2] which are highly relevant to the proposed approach. In particular, [1] is similar to D-Flow, and [2] propose a related approach (although the proposed method is different), that aims to optimize locally the state of the trajectory.
Furthermore, I believe that a comparison with [2] would strenghten the paper, although I understand that for the rebuttal it might challenging.
- b) **Diffusion-based methods:** Although the paper focuses on flow-based methods, the paper misses several methods (almost all of them) from the diffusion literature, which are very related. While citing all of them might be impossible, I suggest the authors to cite at least the surve [3].
- c) **Specific Comparisons:** Missing reference to FLAIR [4], which is a relevant baseline in this domain.
- d) **Decoupling Literature:** The idea of decoupling data consistency from the prior has been considered in several previous diffusion-based works, **DiffPIR** [5] , **RLSD** [6], **Split Gibbs sampling-based methods** [7,8], and **Resample** [9]. A discussion on how MS-Flow’s "Trajectory Consistency" differs fundamentally from these proximal/splitting approaches would strengthen the paper. Clearly, those methods arrive to the idea of decoupling data and prior from a different path, and also some of them are tailord to latent diffusion models, but a discussion would strenghten the work.


2. **Analysis of the Jacobian-Free (JF) Approximation**
The authors highlight the Jacobian-free (JF) update as a key component for improving efficiency (as supported by Table 2). Could a similar JF approximation be applied to the single-shooting D-Flow baseline? If so, how would D-Flow+JF perform in practice? A direct comparison would help clarify whether the improvements primarily arise from the multiple-shooting formulation of MS-Flow or from the use of the Jacobian-free approximation itself. If D-Flow+JF performs worse, this would further support the claim that the short-segment structure of MS-Flow is what enables the JF approximation to work effectively.

3. **Experimental Details and Evaluation**
* **Sample Size:** The evaluation uses 50 images for averaging results. This is significantly lower than the community standard (typically 100–1000 images), which is necessary to ensure the statistical significance of the PSNR/SSIM improvements.
This is even worse for the latent space, which uses only 10 images; given that scalability to latent models is presented as a contribution, the experimental results should be done over a larger validation set.
* **Missing Metrics:** Can the authors include additional metrics, like **FID** and **LPIPS**, which are used in assessing the performance of the solvers.


4. **Difficulty of Tasks**
* The **Super-resolution x2** task is relatively "easy" for modern generative priors. Testing the method on **Super-resolution x4** would provide a better benchmark for the method's ability to handle more challenging and ill-posed problems.
* Furthermore, the paper lacks experiments on non-linear inverse problems, which would strengh the paper's contribution; for instance, DMPlug consider non-linear inverse problems.

5. **Hyperparameter analysis**
In the experiments, D-Flow appears to use $n_t = 3$, while MS-Flow uses $n_t = 6$ to $12$.
* Is this difference intended to match total running time?
* If MS-Flow is given significantly more timesteps, the comparison may be biased. A plot showing the **PSNR vs. Inference Time** for both methods would be informative to understand and compare both methods. In particular, the time-complexity of MS-flow is worse than D-flow for L > 1 (which is the case for almost all the inverse problems).


## Minor Points/questions
* Please provide a direct reference to the hyperparameters used for each experiment (potentially in an Appendix table) within the main experimental section.

* Recent work in latent models (e.g., PSLD, RLSD) has shown that solving inverse problems can sometimes lead to blurry results or specific artifacts. Given that MS-Flow uses an auxiliary variable $x_*$ that can deviate from the exact ODE solution, have the authors observed any such artifacts?

[1] Wang, H., Zhang, X., Li, T., Wan, Y., Chen, T., & Sun, J. (2024). Dmplug: A plug-in method for solving inverse problems with diffusion models. NeurIPS.

[2] Zheng, Yang, Wen Li, and Zhaoqiang Liu. "Integrating intermediate layer optimization and projected gradient descent for solving inverse problems with diffusion models." arXiv preprint arXiv:2505.20789 (2025).

[3] Daras, Giannis, Hyungjin Chung, Chieh-Hsin Lai, Yuki Mitsufuji, Jong Chul Ye, Peyman Milanfar, Alexandros G. Dimakis, and Mauricio Delbracio. "A survey on diffusion models for inverse problems." arXiv preprint arXiv:2410.00083 (2024).

[4] Erbach, Julius, Dominik Narnhofer, Andreas Dombos, Bernt Schiele, Jan Eric Lenssen, and Konrad Schindler. "Solving inverse problems with flair." NeurIPS 2025.

[5] Zhu, Y., Zhang, K., Liang, J., Cao, J., Wen, B., Timofte, R., & Van Gool, L. (2023). Denoising diffusion models for plug-and-play image restoration. In Proceedings of the IEEE/CVF conference on computer vision and pattern recognition (pp. 1219-1229).

[6] Zilberstein, N., Mardani, M., & Segarra, S. (2024). Repulsive latent score distillation for solving inverse problems. ICLR 2025.

[7] Faye, Elhadji C., Mame Diarra Fall, and Nicolas Dobigeon. "Regularization by denoising: Bayesian model and Langevin-within-split Gibbs sampling." IEEE Transactions on Image Processing 34 (2024): 221-234.

[8] Wu, Z., Sun, Y., Chen, Y., Zhang, B., Yue, Y., & Bouman, K. L. (2024). Principled probabilistic imaging using diffusion models as plug-and-play priors. Advances in Neural Information Processing Systems, 37, 118389-118427.

[9] Song, B., Kwon, S. M., Zhang, Z., Hu, X., Qu, Q., & Shen, L. Solving inverse problems with latent diffusion models via hard data consistency. ICLR 2024

[10] Rout, L., Raoof, N., Daras, G., Caramanis, C., Dimakis, A., & Shakkottai, S. (2023). Solving linear inverse problems provably via posterior sampling with latent diffusion models. NeurIPS.

---

> ### Author Rebuttal · Authors · 2026-03-31
>
> We thank you for your careful review. We appreciate that you highlighted the novelty of the multiple-shooting formulation, the constant-memory scaling, the theoretical analysis, the empirical advantages, and the overall clarity of the paper. We added a broader related-work discussion, clarified the novelty relative to closely related diffusion-based approaches, expanded the evaluation to larger test sets (100 images for all settings), added LPIPS, added harder tasks including $4\times$ super-resolution and non-linear deblurring. We hope that you find our responses satisfactory, and that you will consider revising your score.
>
> **Related work and novelty relative to DMILO**
>
> Thank you for pointing us to these references. We added discussion of DMPlug [1], DMILO [2], the survey [3], FLAIR [4], and broader decoupling literature.
>
> Regarding **DMILO [2]**, while both methods go beyond optimizing a single initial latent, they differ in a fundamental way:
>
> | Aspect | DMILO | MS-Flow (Ours) |
> |---|---|---|
> | Optimization variables | *Sequential* optimization of intermediate DDIM-layer inputs | *Joint* optimization of full trajectory |
> | Dynamics enforcement | Solution must follow the hard sampling chain | Uses soft trajectory-stitching penalties |
> | Data consistency | Local in the optimization chain | Separate update of $x_*$ |
> | Backpropagation | Through diffusion sampling step | Avoided in the Jacobian-free variant |
>
> The key novelty of MS-Flow is the multiple-shooting formulation, enabling joint optimization, soft dynamics enforcement, and decoupled terminal updates, which lead to improved conditioning and memory efficiency. We added this clarification to the revised paper.
>
> **Decoupling literature**
>
> We added a broader discussion of diffusion-based decoupling methods. In DiffPIR, RLSD, Split Gibbs-type methods, and Resample, decoupling typically separates prior and data-consistency steps. In MS-Flow, the term
>
> $$
> \sum_{k=1}^K \|x_k - F_{k-1,k}(x_{k-1})\|^2
> $$
>
> acts as a **soft dynamics penalty** over lifted trajectory variables. This formulation improve conditioning, permits local enforcement of the dynamics, and makes the Jacobian-free approximation (JFB) effective over short segments.
>
> While there is a connection to splitting-based inverse-problem solvers at a high level, the main contribution here is the **multiple-shooting-based optimization of a flow trajectory**, rather than only the general idea of decoupling data consistency from a prior.
>
> **FLAIR**
>
> We added a comparison in the latent-flow setting (see *response to Reviewer ebcg*).
>
> **Regarding D-Flow+JFB**
>
> Thank you for this important question. To clarify whether the gains come from multiple shooting or only from the Jacobian-free approximation, we evaluated a D-Flow + JFB variant obtained by replacing the terminal Jacobian
> $$
> \frac{d x(1; x_0)}{d x_0}
> $$
> with the identity.
>
> This performed very poorly with a drop of more than 10 dB in PSNR, together with unrealistic reconstructions. We added this comparison and discussion to the revised paper.
>
> This supports our claim that the benefit does not come merely from using a JFB update. In single shooting, the JFB approximation is applied over the **entire trajectory** from $t=0$ to $t=1$, which is too crude in practice. In **MS-Flow**, the same approximation is used only **locally over short segments**, where it is much more accurate and stable.
>
> **Harder tasks and non-linear inverse problems**
>
> We added both **$4\times$ super-resolution** and **non-linear deblurring** to the revised paper. Please see the response to Reviewer 9cBy. Note, that the latent SD 3.5 experiment is already nonlinear, since the data-consistency update is performed through the nonlinear decoder.
>
> **Regarding NFE fairness and PSNR vs. inference time**
>
> We clarified the NFE accounting in the revised paper. A key subtlety is that **D-Flow uses the midpoint rule**, so $n_t = 3$ corresponds to **6 NFEs**, whereas in **MS-Flow** with explicit Euler, $n_t = 6$ also corresponds to **6 NFEs**.
> We also added a **PSNR vs. inference-time plot** to the revised paper.
>
> More broadly, we clarified the efficiency claim as follows: the main advantage of **MS-Flow** is not that it has a strictly better asymptotic time complexity in every regime, especially when $L > 1$, but that it combines:
> 1. **constant-memory scaling** with respect to trajectory length,
> 2. **improved conditioning** through multiple shooting,
> 3. a practical **Jacobian-free implementation**.
>
> We also added a direct reference to the hyperparameter table.
>
> **Regarding possible artifacts**
>
> We did not observe systematic blur or artifacts indicating that the auxiliary variable $x_*$ drifts too far from the flow manifold. This is controlled by the coupling term
> $$
> \frac{\alpha}{2}\|x_* - x_K\|^2,
> $$
> which keeps the terminal state close to the flow trajectory while still allowing the data-consistency step to improve the inverse solution. We added this clarification to the revised paper.

---

> > ### Author Rebuttal · Reviewer_uhNR · 2026-03-31
> >
> > Thank you for the detailed response! All my concerns were addressed, so I increased my score accordingly, and I lean towards acceptance. Still, the results on more challenging experiments are not encouraging. Given performance and running time, MS-Flow does not outperform the baselines: in terms of performance, it is worse than D-flow but faster, and in terms of running time, it is still slower than PnP methods. This is a weakness that deserves further experiments to justify the method. Therefore, I am not increasing it to 5 at this point.
> >
> > I will wait until the end of the discussion period to allow authors to present an experiment that illustrates the advantages of MS-Flow in more challenging tasks.

---

> > > ### Author Response · Authors · 2026-04-02
> > >
> > > We thank you for the thoughtful follow-up, and for increasing your score reflecting your positive assessment of our paper. We agree that performance on more challenging experiments is important.
> > >
> > > Therefore, and inspired by your review and follow-up comments, we now add two experiments: (i) Phase Retrieval (a non-linear inverse problem, following the settings in [1]); and (ii) out-of-distribution performance evaluation. \
> > > Below, we provide the results and a discussion on these experiments, which we have also added to our revised paper.
> > >
> > >
> > > *(i) Phase Rerieval experiment:*
> > > We evaluated the Fourier Phase Retrieval non-linear inverse problem as in [1] on the same CelebA test set from our main paper. As can be seen from the Table below, MS-Flow achieves a higher SSIM and LPIPS compared to D-Flow, while requiring about half the computation time of D-Flow. This experiment further highlights the contribution of MS-Flow on non-linear inverse problems.
> > >
> > > | CelebA | PSNR ($\uparrow$) | SSIM ($\uparrow$) | LPIPS ($\downarrow$) | Runtime [sec per img] ($\downarrow$) |
> > > | -------- | -------- | -------- |-------- |-------- |
> > > | D-Flow      | 25.12 | 0.656 | 0.193  | 93   |
> > > | MS-Flow (Ours) | 24.08 | 0.691 | 0.187  | 58   |
> > >
> > > (ii) Out-of-distribution experiment:
> > > As another challenging setting, we highlight the advantage of MS-Flow in an out-of-distribution (OOD) scenario.
> > > D-Flow optimizes only the source/noise point by differentiating through the pretrained flow, and is therefore tightly coupled to a single flow trajectory. While this provides a strong prior in-distribution, prior work on generative priors shows that exact restriction to a learned generator can be limiting in inverse problems, especially under distribution shift [2]. In contrast, MS-Flow enforces adherence to the flow trajectory through the trajectory loss while allowing optimization over intermediate states. We therefore expect it to be more robust on OOD data. \
> > > To test this hypothesis, we apply the pretrained CelebA flow model from experiment (i) to the AFHQ-cat dataset for the phase retrieval task. The results are reported in the table below. MS-Flow consistently outperforms D-Flow, achieving better PSNR, SSIM, and LPIPS, while also reducing runtime.
> > >
> > >
> > > | AFHQ-cat | PSNR ($\uparrow$) | SSIM ($\uparrow$) | LPIPS ($\downarrow$) | Runtime [sec per img] ($\downarrow$) |
> > > | -------- | -------- | -------- |-------- |-------- |
> > > | D-Flow   | 19.33  |0.409   |0.502   | 93 |
> > > | MS-Flow (Ours) | 23.70  | 0.689  | 0.269  | 58 |
> > >
> > >
> > > Reference:
> > >
> > > [1] Chung, Hyungjin, et al. "Diffusion Posterior Sampling for General Noisy Inverse Problems." The Eleventh International Conference on Learning Representations (ICLR), 2023.
> > >
> > > [2] Duff, Margaret, et al. "Regularising Inverse Problems with Generative Machine Learning Models". Journal of Mathematical Imaging and Vision, 2021.
> > >
> > > ---
> > >
> > > We would like to conclude by thanking you for the engagement and inspiring follow-up comment. We believe that the added experiments further highlight the merit and effectiveness of our MS-Flow, and that your comments helped us to improve the quality of our paper. \
> > > We would like to kindly ask you to consider revising your score accordingly, as mentioned in your follow-up comment.
> > >
> > > With kindest regards, \
> > > The Authors.

---

### Official Review · Reviewer_6AMv · 2026-03-11

**Soundness:** 3
**Presentation:** 3
**Significance:** 3
**Originality:** 3
**Overall Recommendation:** 5
**Confidence:** 4

**Summary:**

The paper proposes MS-Flow, a training-free inverse problem solver based on pre-training flow models, with the aim of solving the poor conditioning and exploding memory footprint of the D-Flow algorithm. Instead of representing the whole trajectory using only one set of variables at the start and forcing everything else to follow from that, it introduces extra "checkpoints" along the trajectory that are treated as variables too. The method then encourages these checkpoints to agree with what the flow dynamics would produce between them, using a penalty that measures mismatch. The idea proposed is also reminiscent of the principle behind consistency models.

**Compliance With Llm Reviewing Policy:**

Affirmed.

**Final Justification:**

The authors have adequately responded to my concerns. I have raised my score by one point.

**Key Questions For Authors:**

The optimization viewpoint is useful algorithmically, but I am not convinced it is the right or only explanatory lens. Because MS-Flow introduces auxiliary variables, quadratic stitching penalties, and decoupled subproblems, it looks very close to a splitting-style or variational posterior approximation rather than a fundamentally different inference principle. Can the authors discuss whether their method can be interpreted as a variational inference approximation on distributions defined in the joint space? I believe that this viewpoint maybe more fruitul to extend the method to SDE-based models.

**Limitations:**

The main limitation of the paper is the runtime and memory footprint. It remains to be addressed by the authors.

**Strengths And Weaknesses:**

The main idea of the paper is technically principled and intuitive. The authors also clearly demonstrate that their method has significantly improved the stability of the D-Flow. The presentation is also clear and simple to follow.

Regarding the weaknesses, while the paper details the number of shooting points and steps used, it does not provide matched NFE and wall-clock comparisons against the more recent solvers. For example FlowDPS can be quite fast in practice and here it is not clear if the existing methods has any edge at equal computational cost. Can the authors clarify this matter and report the runtime/memory usage?

---

> ### Author Rebuttal · Authors · 2026-03-31
>
> We thank you for your thoughtful feedback. We are grateful that you found the main idea technically principled and intuitive, that you viewed the stability improvements over D-Flow as clearly demonstrated. Your comments helped us sharpen both the conceptual interpretation of MS-Flow and the empirical presentation. Following your suggestions, we added matched runtime comparisons, clarified the NFEs, and expanded the discussion of the method’s probabilistic interpretation in the revised paper. We hope that you find our responses satisfactory, and that you will consider revising your score.
>
> **Joint-posterior interpretation**
>
>  We agree that the optimization viewpoint is not the only useful lens, and that a complementary interpretation in a lifted joint space is natural for MS-Flow. A useful complementary view is that our objective in Eqn. (7) can be interpreted as **MAP estimation over the full trajectory** $x_{0:K}$ together with the terminal reconstruction variable $x_*$. In particular, consider the joint density
> $$
> p(x_*, x_{0:K} \mid y)
> \propto
> \exp \big(-\Phi(x_*)\big)
> \exp \left(-\frac{\alpha}{2}\|x_* - x_K\|^2\right)
> \exp \big(-\lambda R(x_0)\big)
> \prod_{k=1}^K
> \exp \left(-\frac{\gamma}{2}\|x_k - F_{k-1,k}(x_{k-1})\|^2\right).
> $$
> Taking the negative log gives exactly the MS-Flow objective:
> $$ \Phi(x_*) + \frac{\alpha}{2}\|x_* - x_K\|^2 + \lambda R(x_0) + \frac{\gamma}{2}\sum_{k=1}^K \|x_k - F_{k-1,k}(x_{k-1})\|^2.$$
>
> The trajectory-consistency terms act as local Gaussian transition penalties around the discretized dynamics, the $x_*$ coupling is a soft terminal consistency, and the prior on $x_0$ is the latent regularizer. So we agree that MS-Flow can indeed be viewed as a **splitting-based lifted posterior / penalized MAP formulation** in the joint trajectory space.
>
> We would like to clarify the contribution more precisely. Our claim is not that MS-Flow introduces a fundamentally different inference principle from other splitting-based methods. Rather, the novelty is the **multiple-shooting-based formulation**, which yields the concrete advantages studied in our work: local enforcement of dynamics, improved conditioning, constant-memory scaling with respect to the trajectory length. We added this clarification to the revised paper.
>
> We also agree that this viewpoint is promising for extensions to score-based models. Here, the local transitions would be replaced by Euler-Maruyama steps, leading to an similar lifted posterior over intermediate states. We added this discussion to the revised paper.
>
> **Runtime, NFE, and memory**
>
> We agree that matched compute comparisons are essential, especially relative to recent solvers. To address this directly, we now additionally report practical runtime comparisons for the image-recovery experiments on CelebA:
>
> | Method | Time per image [s] |
> |---|---:|
> | Flow-Prior | 24.3 |
> | OT-ODE | 5.19 |
> | PnP-Flow | 5.97 |
> | D-Flow | 26.3 |
> | **MS-Flow (Ours)** | 14.1 |
>
> These results show that on our CelebA restoration setup, **MS-Flow** is substantially faster than **D-Flow**, while also yielding better reconstruction quality.
>
> We also clarified the NFE accounting in the revised paper. One important subtlety is that **D-Flow uses the midpoint rule**, so $n_t = 3$ corresponds to **6 NFEs**, whereas in **MS-Flow** with explicit Euler, $n_t = 6$ also corresponds to **6 NFEs**.
>
> For the latent-flow setting, we also provide a direct runtime / memory comparison below:
>
> | FFHQ Deblurring |  PSNR ($\sigma_{\text{noise}} = 0.01$)  | Time per image [s] | Peak memory [GB] |
> |---|---:|---:|---:|
> | FlowDPS | 27.78 |  16.45 | 8.234 |
> | FLAIR | 29.18 | 10.90 | 16.312 |
> | **MS-Flow (Ours)** | 30.65 |  152.60 | 10.947 |
>
> More broadly, we clarified the efficiency claim in the revised paper. Our point is not that **MS-Flow** dominates every solver in raw wall-clock time in every regime. Rather, the main practical advantage is the combination of **improved conditioning**, **constant-memory scaling**, and strong empirical performance. In practice, it is already faster than D-Flow on our restoration setup.
>
> The difference in Peak GPU memory depends on the specific implementation (i.e., text encoder can be offloaded once the prompt is encoded). In theory all three methods require similar GPU memory, i.e., one forward pass of the model without backpropagation.
>
> **Regarding the main limitation**
>
> Runtime and memory are the important practical considerations. Memory is the main bottleneck for single-shooting methods such as D-Flow, and this is precisely where MS-Flow provides an advantage through constant-memory scaling with respect to the trajectory length. Runtime remains task- and implementation-dependent, but the new wall-clock numbers above show that the method is practical in the regimes we evaluate.
>
> We thank you again for these valuable suggestions. They helped us sharpen both the conceptual positioning of MS-Flow and the empirical presentation of its efficiency.

---

> > ### Author Rebuttal · Reviewer_6AMv · 2026-04-04
> >
> > I would like to thank the reviewers for their rebuttal. I appreciate the joint-space intepretation. Regarding the comparison with Flow DPS and FLAIR and I believe the final paper should contain more thorough comparison with these methods so as to clarify what advantage this method bears.

---

> > > ### Author Response · Authors · 2026-04-07
> > >
> > > Thank you for your response and for taking the time to carefully review our rebuttal.
> > >
> > > We are glad that the joint-space interpretation was helpful and addressed your initial questions. We actually had not considered this interpretation before your review, and we agree it is a very interesting point and provides a promising direction for extending the method to other generative models in the future.
> > >
> > > **Regarding FlowDPS and FLAIR:** following your review, we have provided experimental results, comparing our MS-Flow with FlowDPS and FLAIR, thoroughly discussed in our response. Furthermore, following your current comment, we have expanded the discussion on this in our revised paper. Concretely, we see the three related methods as representing distinct conceptual viewpoints on the problem:
> > > - (i) FlowDPS: Tweedie-like guidance.
> > > - (ii) FLAIR: Variational regularization with a regularizer specified by the flow model.
> > > - (iii) MS-Flow (Ours): Global optimization over the full trajectory.
> > >
> > > We therefore included an expanded comparison (including reconstruction time and  peak memory cost, as reported in our previous response) with these methods in the main text of the final camera-ready version. We believe that these additions further help to highlight the contribution and novelty of MS-Flow, and to distinguish it from existing literature.
> > >
> > >
> > > ---
> > >
> > >
> > > We thank you for the positive assessment of our work, and for acknowledging that you are satisfied with our rebuttal, which has fully resolved your concerns. We believe that your comments helped us improve the quality of our work.
> > >
> > > We would be grateful if following our response, you would consider revising your score accordingly.
> > >
> > >
> > > Thank you, and with kindest regards,
> > >
> > > The Authors.

---

### Official Review · Reviewer_ebcg · 2026-03-12

**Soundness:** 3
**Presentation:** 3
**Significance:** 3
**Originality:** 3
**Overall Recommendation:** 4
**Confidence:** 4

**Summary:**

Authors introduce MS-Flow to solve inverse problems with flow-based models. Instead of optimizing the initial latent noise realization as is done conventionally, authors repesesent the reverse diffusion trajectory as a sequence of intermediate latent states. MS-Flow alternates between data consistency updates and updating intermediate latent states via trajectory-matching penalties. By doing so, authors reduce memory consumption and improve reconstruction quality. Effectiveness of MS-Flow is demonstrated over diverse set of inverse problems.

**Compliance With Llm Reviewing Policy:**

Affirmed.

**Final Justification:**

In the rebuttal period, the authors added additional evaluations (e.g., perceptual metrics, FLAIR comparison, and non-linear forward model experiments), which help better contextualize the results and provide a more complete picture.

In the second round, the authors further strengthened the empirical section by including experiments on OOD generalization and salt-and-pepper noise, where MS-Flow shows improvements over D-Flow. While these results are encouraging, I believe my overall assessment remains unchanged. On more challenging tasks, the conclusions are still somewhat mixed, and the method does not consistently outperform alternatives. It would be beneficial to clearly acknowledge these limitations in the final version.

That said, I still consider this a good paper. The idea is original and theoretically well-justified, and I maintain my positive outlook, increasing my confidence to $4$.

**Key Questions For Authors:**

* In terms of evaluation metrics, providing perceptual quality scores such as LPIPS and FID would be good to complement the distortion metrics (PSNR/SSIM) and provide a more comprehensive view (as they are known to be at odds with each other [2]).
* Authors use linear inverse problems as a test-bed. How about non-linear tasks? Would MS-Flow translate well there as well?
* Included below FLAIR [1], a recent and relevant paper on solving inverse problems with flow models. I would recommend including it for better contextualization of the work.
* In Figure 2, shaded area is very difficult to see on my screen. I would recommend adjusting it for clarity.
* How does MS-Flow compare to baselines in terms of sampling speed (in practice)?

***
***References:***

[1] Erbach, Julius, et al. "Solving inverse problems with flair." arXiv preprint arXiv:2506.02680 (2025).

[2] Blau, Yochai, and Tomer Michaeli. "The perception-distortion tradeoff." Proceedings of the IEEE conference on computer vision and pattern recognition. 2018.

**Limitations:**

yes

**Strengths And Weaknesses:**

***Strengths:***
* The idea of trajectory stitching in the context of inverse problems is an original contribution to the best of my knowledge. The proposed method is theoretically justified well as well.
* Experiments and ablation studies are comprehensive. Furthermore, improvements in terms of performance compared to D-Flow is significant.
* The paper is written very well and it is easy to follow.

***
***Weaknesses:***
* Authors report only distortion metrics for reconstruction performance. Reporting perceptual image quality metrics would complement it.
* Problem setting seem to be restricted to linear inverse problems.
* Please see the questions below.

---

> ### Author Rebuttal · Authors · 2026-03-31
>
> We thank you for the constructive feedback. We are grateful that you found the idea of trajectory stitching original, the theoretical development well-justified, the experiments comprehensive, and the presentation clear and easy to follow. We also appreciate your suggestions for strengthening the empirical evaluation. Following your comments, we have expanded the evaluation and will revise the manuscript accordingly. We hope that you find our responses satisfactory, and that you will consider revising your score.
>
> **Regarding perceptual metrics and practical runtime**
>
> We agree that perceptual metrics complement distortion metrics. We now additionally report **LPIPS** for the CelebA image recovery experiments, alongside the original **PSNR/SSIM** results from the paper, and we also report the **runtime per image**. We further increased the evaluation set from 50 to 100 images. For reference, the CelebA image-recovery results in the paper are reported on **$128 \times 128$** images.
>
> | Method | Gaussian Debl. PSNR | Gaussian Debl. SSIM | Gaussian Debl. LPIPS | Box-Inpainting PSNR | Box-Inp. SSIM | Box-Inp. LPIPS | SR PSNR |SR SSIM | SR LPIPS | Time per image [s] |
> |---|---:|---:|---:|---:|---:|---:|---:|---:|---:|---:|
> | Flow-Prior | 36.12 | 0.961 | 0.009 | 36.79 | 0.984 | 0.011 | 34.47 | 0.946 | 0.015 | 24.3 |
> | OT-ODE | 37.51 | 0.969 | 0.015 | 34.26 | 0.975 | 0.013 | 35.21 | 0.956 | 0.008 | 5.19 |
> | PnP-Flow | 36.19 | 0.962 | 0.049 | 36.43 | 0.985 | 0.011 | 34.73 | 0.948 | 0.059 | 5.97 |
> | D-Flow | 33.17 | 0.890 | 0.029 | 33.96 | 0.979 | 0.012 | 34.36 | 0.942 | 0.019 | 26.3 |
> | **MS-Flow (Ours)** | **38.02** | 0.958 | **0.009** | 36.30 | 0.980 | 0.015 | **36.46** | **0.957** | 0.023 | 14.1 |
>
> These additional results show that **MS-Flow** remains strongest on the distortion-oriented metrics targeted by our objective, while LPIPS is mixed on some tasks. We believe this is consistent with the known perception-distortion tradeoff, and we will explicitly discuss this in the revised manuscript.
>
> **Non-linear inverse problems**
>
> Thank you for this important question. While our main image-recovery experiments use linear forward operators, the method itself is not restricted to this setting. In particular, our **latent-flow experiment with SD 3.5 is already nonlinear**, since the data-consistency term is imposed in image space through the nonlinear decoder $D$, and the corresponding update is solved by gradient descent rather than in closed form.
>
> In addition, following your suggestion, we now include an explicit **non-linear deblurring** benchmark. The corresponding results are reported below and will be added to the revised manuscript.
>
> | Non-linear Deblurring |  PSNR |  SSIM |  LPIPS |
> |---|---:|---:|---:|
> | Flow-Prior | 21.84 | 0.490  |  0.318  |
> | OT-ODE | 21.95 | 0.626 | 0.118 |
> | PnP-Flow | 22.19  | 0.643  | 0.263 |
> | D-Flow | 26.41 |  0.682 | 0.069 |
> | **MS-Flow (Ours)** | 21.93 | 0.606 | 0.284 |
>
> We will also clarify this point in the paper by making more explicit that Equation (10b) is not tied to the closed-form linear case, and that the latent-model setting already demonstrates applicability beyond linear pixel-space inverse problems.
>
> **FLAIR and broader contextualization**
>
> We are very grateful for the pointer to **FLAIR**. We agree that it is relevant and should be discussed in the paper. We now include a comparison against **FLAIR** in the latent-flow setting. The results are reported below and will be incorporated into the revised manuscript together with an expanded discussion in related work.
>
> | FFHQ Deblurring |  PSNR ($\sigma_{\text{noise}} = 0.01$) |  SSIM ($\sigma_{\text{noise}} = 0.01$) | PSNR ($\sigma_{\text{noise}} = 0.1$) | SSIM ($\sigma_{\text{noise}} = 0.1$) |
> |---|---:|---:|---:|---:|
> | FlowDPS | 27.78 | 0.751 | 26.87 | 0.697 |
> | FLAIR | 29.18 | 0.783 | 28.93 | 0.775 |
> | **MS-Flow (Ours)** | 30.65 | 0.837 | 28.62 |  0.772 |
>
> **Figure 2 clarity**
>
> We agree that the current shaded regions in Figure 2 can be made easier to read, and we will revise the figure in the revised manuscript.
>
> **Sampling speed in practice**
>
> Thank you for this important question. We agree that practical speed is important in addition to memory. As shown in the runtime column of the table above, **MS-Flow** is substantially faster than **D-Flow** on our CelebA restoration setup (**14.1s** vs **26.3s** per image), while also yielding better reconstruction quality.
>
> In addition, the current paper already shows that **MS-Flow** with the Jacobian-free approximation scales much more gracefully than D-Flow as the number of discretization points increases. For example, D-Flow grows from **7.43s** at $n_t = 3$ to **175.57s** at $n_t = 12$, whereas MS-Flow with JFB and $L = 1$ grows from **2.44s** to **4.44s** over the same range. We will add the practical runtime comparisons above to the revised manuscript as well. Please see https://bashify.io/i/ts0x2Q for a comparison of PSNR vs. execution time.

---

> > ### Author Rebuttal · Reviewer_ebcg · 2026-04-03
> >
> > I would like to thank the authors for their time and effort in responding and running new experiments in a short time. Specifically, I believe the introduction of perceptual metrics, FLAIR comparison, and non-linear forward model experiments complement the results well and provide the full picture.
> >
> > In general, I agree with other reviewers that on harder tasks (such as noisy ($\sigma=0.1$) FFHQ deblurring vs. FLAIR, and non-linear deblurring task against D-Flow), proposed method MS-Flow is not strictly outperforming them. It would be ideal to mention these limitations of the method in the revised manuscript.
> >
> > That said, i still think it is a good paper. The idea is original, and theoretically justified. For these reasons, I am planning to maintain my positive outlook and increase my confidence.

---

> > > ### Author Response · Authors · 2026-04-07
> > >
> > > We thank the reviewer for their acknowledgment of our rebuttal and for increasing their confidence in the assessment to accept our paper. We are pleased that the additional experiments helped provide a fuller picture. We are also grateful for your feedback and thoughtful comments, which we now address in full, below.
> > >
> > > ### **Regarding difficulty of problems**
> > >
> > > We thank you for the comment. To provide a more complete picture of the "difficulty" spectrum, we provide two thorough experimental results:
> > >
> > > **(i) Out-of-Distribution (OOD) generalization:**   We applied the pretrained CelebA flow model to AFHQ-cat images for a Fourier Phase Retrieval task (following the setting in [1]). The results in the Table below demonstrate that MS-Flow handles distribution shifts significantly better than D-Flow --- further highligitng the contribution of MS-Flow. This results are explained as follows: D-Flow optimizes only the initial noise input, forcing the solution to strictly adhere to the learned trajectories of the flow model. Instead, MS-Flow has the ability to deviate from the flow prior as the stitching penalties allows the model to "bend" the trajectory to satisfy data consistency.  **This result has also been discussed with with Reviewer 9cBy**.
> > >
> > >
> > > | Phase Retrieval | PSNR ($\uparrow$) | SSIM ($\uparrow$) | LPIPS ($\downarrow$) | Runtime [sec per img] ($\downarrow$) |
> > > | -------- | -------- | -------- |-------- |-------- |
> > > | *CelebA:* D-Flow      | 25.12 | 0.656 | 0.193  | 93   |
> > > | *CelebA:* MS-Flow (Ours) | 24.08 | 0.691 | 0.187  | 58   |
> > > | *AFHQ-cat:* D-Flow   | 19.33  |0.409   |0.502   | 93 |
> > > | *AFHQ-cat:* MS-Flow (Ours) | 23.70  | 0.689  | 0.269  | 58 |
> > >
> > >
> > > **(ii) Salt-and-papper noise:** We further explore another type of noise, beyond standard Gaussian noise. Specifically, we consider a salt-and-pepper noise to highlight that MS-Flow can be applied to non-smooth convex objective functions (please see also page 4, line 195-201 in the paper where we discuss such noise models). \
> > > In particular, the noise model reads: $y_{i}^{\delta} = y_{i}$ with probability $1-p$ and $y_{i}^{\delta} = 0$ with probability $p/2$ and $y_{i}^{\delta} = 1$ with probability $p/2$. \
> > > Here, the objective function is given by the $L_1$-Norm instead of the mean-squared-error.
> > >
> > > For the forward operator we evaluated **two settings**: (i) Gaussian blur with varying blur kernel, and (ii) noise level (on both easy and hard settings, as reported below).
> > >
> > > Our results, reported in the Tables below, show that compared with D-Flow, our MS-Flow is better able to adapt to this type of noise model, both for the Easy task and the harder task --- where MS-Flow achieves higher PSNR and SSIM. We added these valuable results to the revised paper, inspired by your comment. Thank you.
> > >
> > >
> > > | Easy (noise level 0.05, blur 1.5, kernel 15)  | PSNR ($\uparrow$) | SSIM ($\uparrow$) | LPIPS ($\downarrow$) |
> > > | -------- | -------- | -------- |-------- |
> > > | D-Flow     |  33.79    |   0.935  |    0.033   |
> > > | MS-Flow (ours)     |  37.45    | 0.965     |   0.016  |
> > >
> > >
> > > | Hard (noise level 0.1, blur 3.0, kernel 21) | PSNR ($\uparrow$) | SSIM ($\uparrow$) | LPIPS ($\downarrow$) |
> > > | -------- | -------- | -------- |-------- |
> > > | D-Flow     | 29.04     |  0.851    | 0.067    |
> > > | MS-Flow (ours)     |  30.34    |  0.854    |  0.081   |
> > >
> > > [1] Chung et al. "Diffusion Posterior Sampling for General Noisy Inverse Problems"
> > >
> > > ---
> > >
> > > We thank you for the positive assessment of our work, and for acknowledging your confidence in our work and its recommendation of acceptance. We believe that your comments helped us improve the quality of our paper.
> > >
> > > We would be grateful if following our response that includes further clarifications and experimental results, you would consider revising your score accordingly.
> > >
> > > Thank you, and with kindest regards,
> > >
> > > The Authors.

---

### Decision · Program_Chairs · 2026-04-30

**Decision:**

Accept (regular)

**Comment:**

This paper proposes MS-Flow, a framework for solving inverse problems with flow-based generative models by representing trajectories as sequences of intermediate latent states. The reviewers found the formulation to be technically principled and original, effectively addressing the memory and numerical stability issues inherent in traditional solvers like D-Flow. Key practical advantages include constant memory scaling with respect to trajectory length and improved conditioning. During the rebuttal, the authors strengthened the empirical section by expanding the evaluation on a larger datasets, adding perceptual metrics, and demonstrating robustness in non-linear and out-of-distribution cases. While some reviewers noted that MS-Flow does not consistently outperform baselines on highly ill-posed or "hard" tasks, they agreed that the method offers a compelling efficiency-performance tradeoff. Given the solid technical contribution and the practical importance for high-resolution image recovery, the paper is recommended for acceptance.